# Matrix Completion with Incomplete
# Side Information via Orthogonal Complement Projection

**Gengshuo Chang** [* 1]   **Wei Zhang** [* 1]   **Lehan Zhang** [1]

## Abstract

Matrix completion aims to recover missing entries in a data matrix using a subset of observed entries. Previous studies show that side information can greatly improve completion accuracy, but most assume perfect side information, which is rarely available in practice. In this paper, we propose an orthogonal complement matrix completion (OCMC) model to address the challenge of matrix completion with incomplete side information. The model leverages the orthogonal complement projection derived from the available side information, generalizing the traditional perfect side information matrix completion to the scenarios with incomplete side information. Moreover, using probably approximately correct (PAC) learning theory, we show that the sample complexity of OCMC model decreases quadratically with the completeness level. To efficiently solve the OCMC model, a linearized Lagrangian algorithm is developed with convergence guarantees. Experimental results show that the proposed OCMC model outperforms state-of-the-art methods on both synthetic data and real-world applications.

## 1. Introduction

Low-rank matrix completion infers the entire data matrix from the partial known elements. This method plays a significant role in various fields, such as recommendation systems (Ramlatchan et al., 2018; Chen & Wang, 2022; Xu et al., 2021), multi-label learning (Xia et al., 2023; Liu et al., 2023; Cabral et al., 2011), wireless channel estimation (Zhang et al., 2018; 2020), and image inpainting (Chen & Ng, 2022; Jia et al., 2022). Traditional matrix completion methods, known as transductive matrix completion, mainly

---

[*]Equal contribution  [1]Department of Information and Communication Engineering, Harbin Institute of Technology, Shenzhen. Correspondence to: Wei Zhang <zhangwei.sz@hit.edu.cn>.

*Proceedings of the 42nd International Conference on Machine Learning*, Vancouver, Canada. PMLR 267, 2025. Copyright 2025 by the author(s).

focus on the low-rank properties of the matrix. Techniques such as nuclear norm minimization(Liang et al., 2022; Li et al., 2022; Yang et al., 2021) and alternating minimization(Jain et al., 2013; Gu et al., 2023) have been developed.

Apart from the low-rank property, studies show that if we have some prior information for the target matrix, i.e., side information, the accuracy can be improved furthermore. Typical forms of side information include knowledge about the column/row spaces (Herbster et al., 2020; Vlachos et al., 2018; Zhang et al., 2019) and information represented by graphs (Elmahdy et al., 2022; Suh & Suh, 2022). For graph-based side information, recent works, such as community detection with stochastic block models (Zhang et al., 2021) and hierarchical similarity graphs (Ahn et al., 2024) have established fundamental limits on sample complexity.

In this paper, we study the side information related to the column and row spaces, which arise in many practical applications. For example, in recommendation system, in addition to a small number of known ratings, we often have access to certain user and item features, which can help refine the ratings (Zhao & Guo, 2017; Han et al., 2019). Early research on side information focused on specific applications, such as collaborative filtering (Menon et al., 2011; Pan et al., 2010) and disease association prediction (Natarajan & Dhillon, 2014), integrating it with non-convex optimization methods. To systematically analyze the problem, researchers generally assume that the side information is "perfect", meaning that the available side information can fully describe the row and column spaces of the desired matrix. Based on this assumption, the problem has been analyzed from both the non-convex (Jain & Dhillon, 2013) and convex (Xu et al., 2013) perspectives, leading to the same conclusion that side information can greatly benefit completion accuracy.

However, the perfect assumption is not always valid. In practice, the completeness of the side information can vary significantly (Chiang et al., 2015; Yang et al., 2020; Lu et al., 2016). This may undermine the effectiveness of methods relying on this assumption, making conclusions non-trivial. For example, in recommendation systems, we may lack complete knowledge of a user's preferences, which could help predict ratings. (Rafailidis & Nanopoulos, 2015; Guo et al., 2019; Pujahari & Sisodia, 2019). Similarly, in multi-

label learning, obtaining complete and precise item features is often challenging (Sun et al., 2021; Liu et al., 2021; Wei et al., 2019). Therefore, it is crucial to explore models that account for the potential incompleteness in side information.

Unfortunately, research on incomplete scenarios remains rather limited. Existing research methods include enhancing the interpretability of models to achieve greater robustness (Lu et al., 2016), which mainly address scenarios where the side information experiences small perturbations but do not directly discuss the model's behavior under incomplete side information. Additionally, some methods address matrix completion by decomposing the target matrix into components and optimizing each component independently (Yang et al., 2020; Chiang et al., 2015). However, these approaches emphasize the properties of the matrix after various decomposition, but ignore the low-rank property of the matrix as a whole. A more detailed description of related work has been presented to Appendix A.

**Main contributions.** The main contributions of the paper are summarized as follows:

- We propose an orthogonal complement matrix completion(OCMC) model under the incomplete side information, generalizing existing methods relying on perfect side information to handle more general scenarios. Specifically, after projecting the target matrix onto four orthogonal subspaces, we observe that the orthogonal complement projection plays a more critical role in matrix completion. Based on these insights, the proposed OCMC model creatively utilizes the low-rank property of the orthogonal complement projection, which can effectively reduce sample complexity.

- Based on the probably approximately correct (PAC) learning theory (Angluin, 1988), we derive that the sample complexity for OCMC model to achieve a completion error of $\epsilon$ is of the order:

$$\min \left\{ O(\mathcal{P}^2 \log N/\epsilon^2), O(\mathcal{X}^2 \sqrt{N}/\epsilon^2) \right\},$$

where $\mathcal{P}$ is linearly related to the incompleteness of the side information, and $\mathcal{X}$ is a constant from the nuclear norm constraint. Thus, the sample complexity of proposed OCMC model decreases quadratically as the completeness level increases.

- To solve the proposed OCMC model, we present an efficient linear ADMM algorithm that reformulates the original problem into multiple subproblems. This algorithm addresses the challenge of the OCMC model, which involves simultaneously optimizing the nuclear norm of the entire matrix and its orthogonal complement projection. Unlike traditional ADMM methods, we approximate the subproblems by retaining only the second-order terms of the nonlinear component, enabling a closed-form solution. Consequently, the

OCMC model can be solved iteratively, significantly enhancing computational efficiency.

For synthetic data experiment, the proposed OCMC model achieves superior completion accuracy, especially in scenarios with highly incomplete side information. For real-world datasets, including movie recommendation and multi-label learning tasks, results show that OCMC consistently outperforms state-of-the-art methods, highlighting its potential for broader applications.

**Notations**: Given a vector $l \in \mathbb{R}^m$, we denote its 2-norm by $\|l\|_2$. Given a matrix $\boldsymbol{H} \in \mathbb{R}^{m \times n}$, we denote its $(i, j)$-th entry by $H_{ij}$, $\|\boldsymbol{H}\|_F$ its Frobenius norm, $\|\boldsymbol{H}\|_*$ its nuclear norm and $\|\boldsymbol{H}\|_2$ for its 2-norm or operation norm. We denote the column space and row space of $\boldsymbol{H}$ by col($\boldsymbol{H}$) and row($\boldsymbol{H}$), respectively.

## 2. Matrix Completion Model with Incomplete Side Information

### 2.1. From Complete to Incomplete Side Information

Let $\boldsymbol{R} \in \mathbb{R}^{m \times n}$ be the rank-$r$ desired matrix, where $r \ll \min\{m, n\}$, $N = \max\{m, n\}$, indicating that $\boldsymbol{R}$ is a typical low-rank matrix. And $\Omega \in \{1, 2, \ldots, m\} \times \{1, 2, \ldots, n\}$ denotes the set of the indices of observed entries. We define the sampling operator $\mathrm{P}_\Omega : \mathbb{R}^{n \times m} \to \mathbb{R}^{n \times m}$ as

$$[\mathrm{P}_\Omega(\boldsymbol{R})]_{ij} = \begin{cases} R_{ij}, & (i, j) \in \Omega \\ 0, & (i, j) \notin \Omega \end{cases}$$

In order to recover the matrix $\boldsymbol{R}$ from the sampling set $\Omega$, the nuclear norm is typically used as a convex relaxation of the rank function (Candes & Recht, 2012):

$$\min_{\boldsymbol{X}} \|\boldsymbol{X}\|_* \quad \text{s.t. } \mathrm{P}_\Omega(\boldsymbol{X}) = \mathrm{P}_\Omega(\boldsymbol{R}). \tag{1}$$

In this problem, all matrices satisfying the sampling conditions form a feasible set. When the observed entries are sufficient (Candes & Recht, 2012), the feasible set is relatively small, making it highly likely that the optimal solution corresponds to the desired matrix. However, as the number of observations decreases, the feasible set grows, and the probability of the optimal solution being the desired matrix decreases. In the extreme case, when there is no observations, all matrices of the appropriate size are in the feasible set, and the unique minimizer is the zero matrix. The schematic diagram is shown in Figure 1.[1]

Based on the above discussion, increasing the number of observations will reduce the size of the feasible set, while side information can further narrow it without the need to increase the number of observed entries. For example, let matrices $\boldsymbol{A} \in \mathbb{R}^{m \times r_A}$ and $\boldsymbol{B} \in \mathbb{R}^{n \times r_B}$ represent the side

---

[1] The figure is generated via online plotting tool https://www.chiplot.online/.

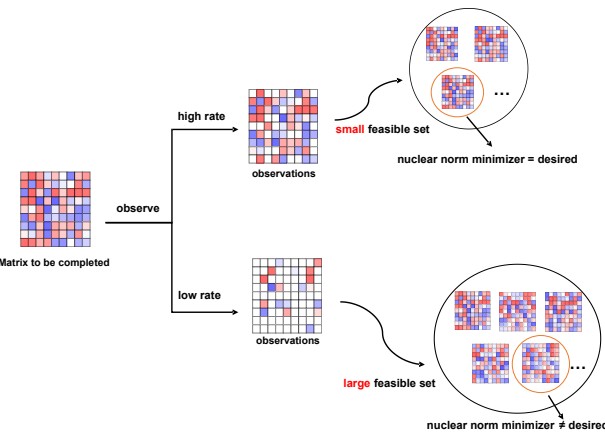

*Figure 1. Matrix completion under different sampling conditions.*

information related to the column and row subspaces of $\boldsymbol{R}$, respectively. In particular, when the side information is perfect, we have

$$\mathrm{col}(\boldsymbol{R}) \subseteq \mathrm{col}(\boldsymbol{A}), \quad \mathrm{row}(\boldsymbol{R}) \subseteq \mathrm{col}(\boldsymbol{B}).$$

Thus, there is a projection representation relationship between $\boldsymbol{R}$ and $\boldsymbol{A}, \boldsymbol{B}$, i.e., $\boldsymbol{R} = \boldsymbol{A}\boldsymbol{M}\boldsymbol{B}^T$, where $\boldsymbol{M}$ can be referred to as the image matrix. In other words, the target matrix $\boldsymbol{R}$ fully lies in the spaces defined by $\boldsymbol{A}$ and $\boldsymbol{B}$. Therefore, we can further reduce the feasible set in (1) by imposing the projection constraint, i.e., $\boldsymbol{X} = \boldsymbol{A}\boldsymbol{M}\boldsymbol{B}^T$,

$$\min_{\boldsymbol{M}} \|\boldsymbol{X}\|_*$$
$$\text{s.t. } \mathrm{P}_\Omega(\boldsymbol{X}) = \mathrm{P}_\Omega(\boldsymbol{R}), \ \boldsymbol{X} = \boldsymbol{A}\boldsymbol{M}\boldsymbol{B}^T. \quad (2)$$

Assuming $\boldsymbol{A}$ and $\boldsymbol{B}$ semi-orthogonal, we have $\|\boldsymbol{M}\|_* = \|\boldsymbol{A}\boldsymbol{M}\boldsymbol{B}^T\|_*$, then (2) can be written as:

$$\min_{\boldsymbol{M}} \|\boldsymbol{M}\|_* \quad \text{s.t. } \mathrm{P}_\Omega(\boldsymbol{A}\boldsymbol{M}\boldsymbol{B}^T) = \mathrm{P}_\Omega(\boldsymbol{R}), \quad (3)$$

leading to the IMC formulation (Xu et al., 2013).

However, when side information is incomplete, the desired matrix can not be constrained to specific projection spaces, as the equality constraint in (2), i.e., $\boldsymbol{X} = \boldsymbol{A}\boldsymbol{M}\boldsymbol{B}^T$, no longer holds. For the purpose of analysis, we introduce the following notations. Let $\boldsymbol{A} \in \mathbb{R}^{m \times r_A}$ and $\boldsymbol{B} \in \mathbb{R}^{n \times r_B}$ denote the available (incomplete) side information, and $\hat{\boldsymbol{A}} \in \mathbb{R}^{m \times \hat{r}_A}$ and $\hat{\boldsymbol{B}} \in \mathbb{R}^{n \times \hat{r}_B}$ denote complete (perfect) side information. We further define $\tilde{\boldsymbol{A}} \in \mathbb{R}^{m \times \tilde{r}_A}$ and $\tilde{\boldsymbol{B}} \in \mathbb{R}^{n \times \tilde{r}_B}$ as the supplementary components associated with the perfect side information. For example,

$$\mathrm{col}(\boldsymbol{R}) \subseteq \mathrm{col}(\hat{\boldsymbol{A}}) = \mathrm{col}(\boldsymbol{A}) \cup \mathrm{col}(\tilde{\boldsymbol{A}})$$
$$\mathrm{row}(\boldsymbol{R}) \subseteq \mathrm{col}(\hat{\boldsymbol{B}}) = \mathrm{col}(\boldsymbol{B}) \cup \mathrm{col}(\tilde{\boldsymbol{B}}),$$

In this work, we define the completeness level of side information as

$$\text{completeness level} = \frac{\mathrm{rank}(\boldsymbol{A}) + \mathrm{rank}(\boldsymbol{B})}{2\,\mathrm{rank}(\boldsymbol{R})}.$$

Without loss of generality, we assume the column spaces of $\boldsymbol{A}$, $\boldsymbol{B}$ and $\tilde{\boldsymbol{A}}$, $\tilde{\boldsymbol{B}}$ have no overlapping, i.e., $\mathrm{col}(\boldsymbol{A}) \cap \mathrm{col}(\tilde{\boldsymbol{A}}) = \varnothing, \mathrm{col}(\boldsymbol{B}) \cap \mathrm{col}(\tilde{\boldsymbol{B}}) = \varnothing$. We also assume that $\boldsymbol{A}$, $\boldsymbol{B}$, $\tilde{\boldsymbol{A}}$ and $\tilde{\boldsymbol{B}}$ are semi-orthogonal matrices, i.e., $\boldsymbol{A}^T \boldsymbol{A} = \boldsymbol{I}$ and similarly for the others.

Based on the definitions above, one has $\hat{\boldsymbol{A}} = [\boldsymbol{A}, \tilde{\boldsymbol{A}}]\boldsymbol{T}_{\boldsymbol{A}}$ and $\hat{\boldsymbol{B}} = [\boldsymbol{B}, \tilde{\boldsymbol{B}}]\boldsymbol{T}_{\boldsymbol{B}}$, where $\boldsymbol{T}_{\boldsymbol{A}}$ and $\boldsymbol{T}_{\boldsymbol{B}}$ refer to the transfer matrices. Then, the target matrix $\boldsymbol{R}$ can be expressed as

$$\boldsymbol{R} = \hat{\boldsymbol{A}}\boldsymbol{M}\hat{\boldsymbol{B}}^T = [\boldsymbol{A}, \tilde{\boldsymbol{A}}]\boldsymbol{T}_{\boldsymbol{A}}\boldsymbol{M}\boldsymbol{T}_{\boldsymbol{B}}[\boldsymbol{B}, \tilde{\boldsymbol{B}}]^T$$
$$= [\boldsymbol{A}, \tilde{\boldsymbol{A}}]\boldsymbol{M}'[\boldsymbol{B}, \tilde{\boldsymbol{B}}]^T.$$

Dividing the matrix $\boldsymbol{M}'$ into four blocks gives

$$\boldsymbol{R} = [\boldsymbol{A}, \tilde{\boldsymbol{A}}] \begin{bmatrix} \boldsymbol{M}^{11} & \boldsymbol{M}^{12} \\ \boldsymbol{M}^{21} & \boldsymbol{M}^{22} \end{bmatrix} [\boldsymbol{B}, \tilde{\boldsymbol{B}}]^T$$
$$= \boldsymbol{A}\boldsymbol{M}^{11}\boldsymbol{B}^T + \boldsymbol{A}\boldsymbol{M}^{12}\tilde{\boldsymbol{B}}^T + \tilde{\boldsymbol{A}}\boldsymbol{M}^{21}\boldsymbol{B}^T + \tilde{\boldsymbol{A}}\boldsymbol{M}^{22}\tilde{\boldsymbol{B}}^T. \quad (4)$$

To recover the target matrix $\boldsymbol{R}$ with incomplete information $\boldsymbol{A}$ and $\boldsymbol{B}$, an intuitive approach is to impose the constraint from (4) on the problem in (1), similar to the formulation in (2). However, since $\tilde{\boldsymbol{A}}$ and $\tilde{\boldsymbol{B}}$ are unavailable, this approach is not valid. Nevertheless, the decomposition form of the target matrix in (4) remains important. Both transductive and inductive matrix completion can be derived from (4). If no side information is available (i.e., $\boldsymbol{A} = \boldsymbol{B} = \varnothing$), the first three terms vanish, and the fourth term becomes the entire matrix $\boldsymbol{R}$, reducing the model to the transaction matrix completion. If perfect side information is available, then the last three terms vanish. Furthermore, existing models for incomplete side information (Chiang et al., 2015; Yang et al., 2020) essentially separate the fully known part from the rest, i.e., the first term from the last three in (4).

## 2.2. Orthogonal Complement Matrix Completion Model

To utilize the decomposition in (4) and derive a valid model for matrix completion with incomplete information, we define the following projection operations $\mathbb{R}^{m \times n} \to \mathbb{R}^{m \times n}$: $\mathrm{P}_{\boldsymbol{A}\boldsymbol{B}}(\cdot)$, $\mathrm{P}_{\boldsymbol{A}\boldsymbol{B}^\perp}(\cdot)$, $\mathrm{P}_{\boldsymbol{A}^\perp\boldsymbol{B}}(\cdot)$, and $\mathrm{P}_{\boldsymbol{A}^\perp\boldsymbol{B}^\perp}(\cdot)$,

$$\mathrm{P}_{\boldsymbol{A}\boldsymbol{B}}(\boldsymbol{X}) = \boldsymbol{P}_{\boldsymbol{A}}\boldsymbol{X}\boldsymbol{P}_{\boldsymbol{B}},$$
$$\mathrm{P}_{\boldsymbol{A}\boldsymbol{B}^\perp}(\boldsymbol{X}) = \boldsymbol{P}_{\boldsymbol{A}}\boldsymbol{X} - \mathrm{P}_{\boldsymbol{A}\boldsymbol{B}}(\boldsymbol{X}),$$
$$\mathrm{P}_{\boldsymbol{A}^\perp\boldsymbol{B}}(\boldsymbol{X}) = \boldsymbol{X}\boldsymbol{P}_{\boldsymbol{B}} - \mathrm{P}_{\boldsymbol{A}\boldsymbol{B}}(\boldsymbol{X}),$$
$$\mathrm{P}_{\boldsymbol{A}^\perp\boldsymbol{B}^\perp}(\boldsymbol{X}) = \boldsymbol{X} - \boldsymbol{P}_{\boldsymbol{A}}\boldsymbol{X} - \boldsymbol{X}\boldsymbol{P}_{\boldsymbol{B}} + \mathrm{P}_{\boldsymbol{A}\boldsymbol{B}}(\boldsymbol{X}),$$

where $\boldsymbol{A}^\perp \in \mathbb{R}^{m \times (m-r_A)}$ and $\boldsymbol{B}^\perp \in \mathbb{R}^{n \times (n-r_B)}$ represent the orthogonal complement matrix (Strang, 2000) of $\boldsymbol{A}$ and $\boldsymbol{B}$ which satisfies: $\mathrm{rank}([\boldsymbol{A}, \boldsymbol{A}^\perp]) = m, \boldsymbol{A}^T\boldsymbol{A}^\perp = \boldsymbol{0}; \mathrm{rank}([\boldsymbol{B}, \boldsymbol{B}^\perp]) = n$ and $\boldsymbol{B}^T\boldsymbol{B}^\perp = \boldsymbol{0}$. Additionally, $\boldsymbol{P}_{\boldsymbol{A}} = \boldsymbol{A}\boldsymbol{A}^T \in \mathbb{R}^{m \times m}$ and $\boldsymbol{P}_{\boldsymbol{B}} = \boldsymbol{B}\boldsymbol{B}^T \in \mathbb{R}^{n \times n}$ represent the projection matrices related to $\boldsymbol{A}$ and $\boldsymbol{B}$. For example, $\mathrm{P}_{\boldsymbol{A}\boldsymbol{B}}(\boldsymbol{X})$ refers to projecting $\boldsymbol{X}$ onto the intersection of column space of $\boldsymbol{A}$ and row space of $\boldsymbol{B}$.

*Table 1.* The range of ranks for different matrix parts.

| PART | MINIMUM RANK | MAXIMUM RANK |
|---|---|---|
| $\mathrm{P}_{\boldsymbol{AB}}(\boldsymbol{X})$ | 0 | $\min\{\mathrm{rank}(\boldsymbol{A}), \mathrm{rank}(\boldsymbol{B})\}$ |
| $\mathrm{P}_{\boldsymbol{AB}^\perp}(\boldsymbol{X})$ | 0 | $\mathrm{rank}(\boldsymbol{A})$ |
| $\mathrm{P}_{\boldsymbol{A}^\perp \boldsymbol{B}}(\boldsymbol{X})$ | 0 | $\mathrm{rank}(\boldsymbol{B})$ |
| $\mathrm{P}_{\boldsymbol{A}^\perp \boldsymbol{B}^\perp}(\boldsymbol{X})$ | 0 | $\min\{m - \mathrm{rank}(\boldsymbol{A}), n - \mathrm{rank}(\boldsymbol{B})\}$ |
| $\mathrm{P}_{\boldsymbol{AB}}(\boldsymbol{X}) + \mathrm{P}_{\boldsymbol{AB}^\perp}(\boldsymbol{X}) + \mathrm{P}_{\boldsymbol{A}^\perp \boldsymbol{B}}(\boldsymbol{X})$ | 0 | $\mathrm{rank}(\boldsymbol{A}) + \mathrm{rank}(\boldsymbol{B})$ |

It is worth noting that target matrix $\boldsymbol{R}$ can be expressed as

$$\boldsymbol{R} = \mathrm{P}_{\boldsymbol{AB}}(\boldsymbol{R}) + \mathrm{P}_{\boldsymbol{A}^\perp \boldsymbol{B}}(\boldsymbol{R}) + \mathrm{P}_{\boldsymbol{AB}^\perp}(\boldsymbol{R}) + \mathrm{P}_{\boldsymbol{A}^\perp \boldsymbol{B}^\perp}(\boldsymbol{R}). \quad (5)$$

Compared to the expression in (4), the form in (5) only involves the incomplete side information $\boldsymbol{A}$ and $\boldsymbol{B}$, which may enable the valid constraint for $\boldsymbol{R}$. However, one can find that any matrix $\boldsymbol{X} \in \mathbb{R}^{m \times n}$ can be expressed as

$$\boldsymbol{X} = \mathrm{P}_{\boldsymbol{AB}}(\boldsymbol{X}) + \mathrm{P}_{\boldsymbol{A}^\perp \boldsymbol{B}}(\boldsymbol{X}) + \mathrm{P}_{\boldsymbol{AB}^\perp}(\boldsymbol{X}) + \mathrm{P}_{\boldsymbol{A}^\perp \boldsymbol{B}^\perp}(\boldsymbol{X}), \quad (6)$$

therefore, directly imposing the constraint from (6) on the problem in (1) is not meaningful.

To address this issue and derive a valid constraint on $\boldsymbol{X}$ from (6), it is necessary to analyze components in (5) and ensure that the corresponding parts of $\boldsymbol{X}$ exhibit properties consistent with (5). Specifically, since the target matrix $\boldsymbol{R}$ is low-rank, one can find that all four components in (5) must also be low-rank. Therefore, for the matrix completion task, the components of candidates $\boldsymbol{X}$ in (6) must maintain the low-rank properties to guide the recovery matrix towards the target $\boldsymbol{R}$. In other words, this requires restricting the feasible set in (1) to the matrices whose components, as defined in (6), are low-rank. Moreover, enforcing the low-rank properties of each component in (6) enables more accurate estimation of target matrix $\boldsymbol{R}$ with fewer observations, as it reduces the number of parameters to be learned. This advantage will be formally justified in Section 3.

In what follows, we will discuss how to impose the low-rank constraints for each part in (6). Interestingly, for any $\boldsymbol{X} \in \mathbb{R}^{m \times n}$, one can note that the first three parts in (6), i.e, $\mathrm{P}_{\boldsymbol{AB}}(\boldsymbol{X})$, $\mathrm{P}_{\boldsymbol{AB}^\perp}(\boldsymbol{X})$, and $\mathrm{P}_{\boldsymbol{A}^\perp \boldsymbol{B}}(\boldsymbol{X})$, inherently satisfy the low-rank property because they include at least one known side information matrix factor. However, the orthogonal complent $\mathrm{P}_{\boldsymbol{A}^\perp \boldsymbol{B}^\perp}(\boldsymbol{X})$ lacks any side information matrix factor, and its low-rank characteristics cannot be guaranteed for any matrix $\boldsymbol{X}$. Therefore, intuitively, it is more important to restrict the rank of the $\mathrm{P}_{\boldsymbol{A}^\perp \boldsymbol{B}^\perp}(\boldsymbol{X})$. To facilitate the discussion, we refer to $\mathrm{P}_{\boldsymbol{A}^\perp \boldsymbol{B}^\perp}(\cdot)$ as the orthogonal complement projection throughout this paper.

Table 1 illustrates the rank ranges of parts in (6) for an arbitrary matrix, highlighting that the special characteristic of $\boldsymbol{R}$ lies in the low-rank property of $\mathrm{P}_{\boldsymbol{A}^\perp \boldsymbol{B}^\perp}(\boldsymbol{R})$. It can be observed that, due to $\mathrm{rank}(\boldsymbol{A}) \leq r_A \ll m$ and $\mathrm{rank}(\boldsymbol{B}) \leq r_B \ll n$, the maximum possible rank of the orthogonal

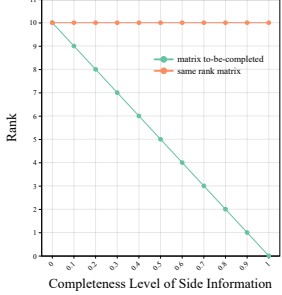 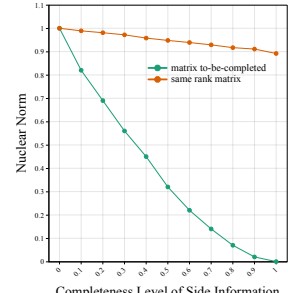

(a) *Rank comparison*      (b) *Nuclear norm comparison*

*Figure 2. We compare the rank and nuclear norm of* $\mathrm{P}_{\boldsymbol{A}^\perp \boldsymbol{B}^\perp}(\boldsymbol{R})$ *and* $\mathrm{P}_{\boldsymbol{A}^\perp \boldsymbol{B}^\perp}(\boldsymbol{X})$, *where* $\boldsymbol{R}$ *is the desired matrix and* $\boldsymbol{X}$ *is a randomly generated matrix in* $\mathbb{R}^{100 \times 100}$ *with the same rank as* $\boldsymbol{R}$.

complement projection $\mathrm{P}_{\boldsymbol{A}^\perp \boldsymbol{B}^\perp}(\boldsymbol{X})$ is significantly greater than even the rest three parts combined. Thus, it is essential to restrict the solution such that $\mathrm{P}_{\boldsymbol{A}^\perp \boldsymbol{B}^\perp}(\boldsymbol{X})$ has a low rank, consistent with the low-rank property of $\mathrm{P}_{\boldsymbol{A}^\perp \boldsymbol{B}^\perp}(\boldsymbol{R})$.

Moreover, as shown in Figure 2, as the completeness of the side information increases, $\mathrm{P}_{\boldsymbol{A}^\perp \boldsymbol{B}^\perp}(\boldsymbol{R})$ and $\mathrm{P}_{\boldsymbol{A}^\perp \boldsymbol{B}^\perp}(\boldsymbol{X})$ exhibit distinct trends. Specifically, both the rank and nuclear norm of $\mathrm{P}_{\boldsymbol{A}^\perp \boldsymbol{B}^\perp}(\boldsymbol{R})$ show the approximately linear decreasing trend, where the rank curve of $\mathrm{P}_{\boldsymbol{A}^\perp \boldsymbol{B}^\perp}(\boldsymbol{R})$ in Figure 2-(a) is straightforward. For the nuclear norm curve of $\mathrm{P}_{\boldsymbol{A}^\perp \boldsymbol{B}^\perp}(\boldsymbol{R})$ in Figure 2-(b), since the projection is constructed based on the subspaces of $\boldsymbol{R}$ itself, the decay approximately at a rate of $1/r$. In contrast, for a same-rank random matrix $\boldsymbol{X}$, the rank of $\mathrm{P}_{\boldsymbol{A}^\perp \boldsymbol{B}^\perp}(\boldsymbol{X})$ remains nearly constant, and its nuclear norm decreases more slowly—at a rate of approximately $1/n$. This is because $\boldsymbol{A}$ and $\boldsymbol{B}$ are derived from the subspaces of the target matrix $\boldsymbol{R}$, rather than from $\boldsymbol{X}$ itself. These empirical observations further highlight the distinctive low-rank property of $\mathrm{P}_{\boldsymbol{A}^\perp \boldsymbol{B}^\perp}(\boldsymbol{R})$.

Based on the discussions above, to perform matrix completion with incomplete information, we use the low-rank property of $\mathrm{P}_{\boldsymbol{A}^\perp \boldsymbol{B}^\perp}(\boldsymbol{X})$ as an additional constraint in (1) to further narrow the feasible set. Since it is not an equality constraint, we incorporate it into the objective function and consider solving the following problem:

$$\min_{\boldsymbol{X}} \; \|\boldsymbol{X}\|_* + \lambda \|\mathrm{P}_{\boldsymbol{A}^\perp \boldsymbol{B}^\perp}(\boldsymbol{X})\|_*$$

$$\text{s.t. } \mathrm{P}_\Omega(\boldsymbol{X}) = \mathrm{P}_\Omega(\boldsymbol{R}). \quad (7)$$

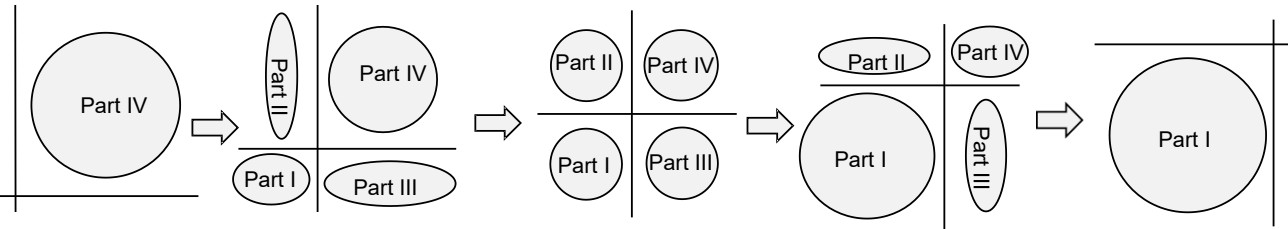

*Figure 3. The trend of components' variation as side information completeness varies. Parts I, II, III, and IV correspond to the first four items of (4). From left to right, the images represent increasing levels of side information completeness. The size of the shaded regions symbolizes the proportion of each component relative to the desired matrix.*

The derived model in (7) is called the orthogonal complement matrix completion (OCMC). In the objective of (7), $\|X\|_*$ and $\|P_{A^\perp B^\perp}(X)\|_*$ are used to incorporate the low-rank property of $X$ and $P_{A^\perp B^\perp}(X)$, accounting for both the global property and the property of orthogonal complement. In addition, it is easy to verify that the problem in (7) is convex, ensuring that it is guaranteed to converge to a global optimum.

**Discussion about $\lambda$:** The parameters $\lambda$ in (7) is used to adjust the weight of $\|P_{A^\perp B^\perp}(X)\|_*$ based on the completeness of the side information. For example, when there is no side information, we have $A = B = 0$, then $\|P_{A^\perp B^\perp}(R)\|_* = \|R\|_*$, which indicates that no additional effects can be achieved because no useful information is included. This scenario can also be interpreted as setting $\lambda \to 0$, and the problem transforms into (1). As the side information becomes more complete, it can be learned from Figure 2 that there is a approximately linear reduction in $\|P_{A^\perp B^\perp}(R)\|_*$, therefore it is reasonable to set a larger $\lambda$ in (7). By adjusting the values of $\lambda$ according to the completeness of the side information, OCMC can effectively incorporate side information into matrix completion. Figure 3 provides a visual representation of the aforementioned trends: as the completeness of side information increases, the proportion of the orthogonal complement projection becomes smaller.

## 3. Recovery Analysis

In this section, we theoretically analyze the recovery accuracy and sample complexity for the proposed OCMC model according to the PAC learning framework.

### 3.1. PAC Model

PAC is a theoretical framework that ensures, with enough samples, an algorithm can produce a model with a small generalization error and high probability (Angluin, 1988). Using this framework, we will show the recovery error of the proposed OCMC model in (7) is bounded.

Considering the potential presence of Gaussian noise in observations, we relax the equality constraint $P_\Omega(X) =$

$P_\Omega(R)$ in (7) by $\ell(X_{ij}, R_{ij})$:

$$\min_X \sum_{(i,j)\in\Omega} \ell(X_{ij}, R_{ij}) + \lambda_1 \|X\|_* + \lambda_2 \|P_{A^\perp B^\perp}(X)\|_*, \quad (8)$$

where the function $\ell(\cdot, \cdot)$ is used to measure the differences at the observed positions. In particular, for the squared loss, one has $\ell(X_{ij}, R_{ij}) = (X_{ij} - R_{ij})^2$. Then problem (8) can be reformulated into the following soft constrained form:

$$\min_X \sum_{(i,j)\in\Omega} \ell(X_{ij}, R_{ij})$$

$$\text{s.t. } \|X\|_* \leq \mathcal{X} \quad \text{and} \quad \|P_{A^\perp B^\perp}(X)\|_* \leq \mathcal{P}, \quad (9)$$

where $\mathcal{X}$ and $\mathcal{P}$ are related to $\lambda_1$ and $\lambda_2$ in (8). Similar to the role of $\lambda$ in (7), the parameter $\mathcal{P}$ in (9) is also related to the completeness level of the side information. As the completeness level increases, we set a smaller value for $\mathcal{P}$. Here, for convenience, we define the feasible set of problem (9) as $\mathfrak{X} = \{X \mid \|X\|_* \leq \mathcal{X}, \|P_{A^\perp B^\perp}(X)\|_* \leq \mathcal{P}\}$.

To relate model (9) to the PAC learning framework, we define a function class $\mathcal{F}$ based on the set $\mathfrak{X}$,

$$\mathcal{F} = \{f_X(i,j) = X_{ij} \mid X \in \mathfrak{X}\}. \quad (10)$$

Based on our definition, one can find there is a one-to-one correspondence between the function $f_X$ and the matrix $X$. Then, we can treat solving problem (9) as finding a function $f_X$ in the function class $\mathcal{F}$, which minimizes the corresponded objective value in (9), i.e.,

$$\min_{f_X \in \mathcal{F}} \sum_{(i,j)\in\Omega} \ell(f_X(i,j), R_{ij}). \quad (11)$$

Moreover, given arbitray $X$, we consider two types of errors: empirical error $\hat{\mathcal{L}}(X)$ and generalization error $\mathcal{L}(X)$,

$$\hat{\mathcal{L}}(X) = \frac{1}{|\Omega|} \sum_{(i,j)\in\Omega} \ell(f_X(i,j), R_{ij}),$$

$$\mathcal{L}(X) = \mathbb{E}_{(i,j)\sim p}[\ell(f_X(i,j), R_{ij})],$$

where $p$ denotes the sampling distribution over the observed entry positions in $\Omega$. Thus, the empirical error calculates the average loss at the observed positions, while the generalization error computes the expected loss over all entries. As we can see, the empirical error is relatively easy to calculate, but the generalization error is more important and

more challenging to obtain. To evaluate the generalization error $\mathcal{L}(\boldsymbol{X})$ based on the empirical error $\hat{\mathcal{L}}(\boldsymbol{X})$, we employ the PAC model (Angluin, 1988), leveraging the Rademacher complexity of function class $\mathcal{F}$.

**Lemma 3.1.** *(Bartlett & Mendelson, 2002) Suppose $\ell(\cdot, \cdot)$ is a loss function with Lipschitz constant $L_\ell$ and bounded by $\mathcal{B}$ with respect to its first argument, and $\delta$ is a constant where $0 < \delta < 1$. Let $\mathfrak{R}_\ell(\mathcal{F})$ be the Rademacher complexity of the function class $\mathcal{F}$ (w.r.t. $\Omega$ and associated with $\ell$):*

$$\mathfrak{R}_\ell(\mathcal{F}) = \mathbb{E}_\sigma \left[ \sup_{\substack{\|\boldsymbol{X}\|_* \leq \mathcal{X} \\ \|\mathrm{P}_{\boldsymbol{A}^\perp \boldsymbol{B}^\perp}(\boldsymbol{X})\|_* \leq \mathcal{P}}} \frac{1}{|\Omega|} \sum_{\alpha=1}^{|\Omega|} \sigma_\alpha l(X_{i_\alpha j_\alpha}, R_{i_\alpha j_\alpha}) \right],$$

*where $\sigma_\alpha$ are random variables taking values of $\{\pm 1\}$ with equal probability. Then with probability at least $1 - \delta$, for all $f \in \mathcal{F}$, i.e., $\forall \boldsymbol{X} \in \mathcal{X}$, the generalization error satisfies*

$$\mathcal{L}(\boldsymbol{X}) \leq \hat{\mathcal{L}}(\boldsymbol{X}) + 2\mathbb{E}_\Omega[\mathfrak{R}_\ell(\mathcal{F})] + \mathcal{B}\sqrt{\frac{\log(1/\delta)}{2|\Omega|}}. \quad (12)$$

From Lemma 3.1, since the last term in (12) is constant, the generalization error is influenced by the empirical error $\hat{\mathcal{L}}(\boldsymbol{X})$ and the Rademacher complexity $\mathbb{E}_\Omega[\mathfrak{R}_\ell(\mathcal{F})]$. By solving problem (9), the empirical error $\hat{\mathcal{L}}(\boldsymbol{X})$ will be a small value. Thus, the most significant factor affecting the generalization error in (12) is the Rademacher complexity of $\mathcal{F}$, which will be discussed in the next subsection.

### 3.2. Main Results

In this part, we focus on analyzing the Rademacher complexity of $\mathcal{F}$ in (10), and establish the bound for $\mathcal{L}(\boldsymbol{X})$ by solving (9). Intuitively, as previously mentioned, the complexity of our function class should be closely related to the structure of the feasible set, i.e., to the parameters $\mathcal{X}$ and $\mathcal{P}$ in Problem (9). Formally, Lemma 3.2 provides an upper bound on the Rademacher complexity.

**Lemma 3.2.** *Let $\boldsymbol{Q} \in \mathbb{R}^{m \times m}$ be a matrix with columns $\boldsymbol{Q} = \{q_i\}$, and $\boldsymbol{S} \in \mathbb{R}^{n \times n}$ be a matrix with columns $\boldsymbol{S} = \{s_i\}$, which make $(\boldsymbol{I} + \mu\boldsymbol{I} - \boldsymbol{P}_{\boldsymbol{B}})\boldsymbol{S} = \boldsymbol{Q}(\boldsymbol{I} + \mu\boldsymbol{I} - \boldsymbol{P}_{\boldsymbol{A}}) = \boldsymbol{I}$ hold for some $\mu > 0$. Then, with the settings in (9), the upper-bound of Rademacher complexity of function class $\mathcal{F}$ in (10), i.e., $\mathbb{E}_\Omega[\mathfrak{R}_\ell(\mathcal{F})]$, is given by*

$$\min \left\{ 2L_\ell \mathcal{Q}\mathcal{S}\mathcal{P}\sqrt{\frac{\log 2N}{|\Omega|}}, \sqrt{\frac{9CL_\ell \mathcal{X}\mathcal{B}(\sqrt{m} + \sqrt{n})}{|\Omega|}} \right\},$$

*where $N = max\{m, n\}$ denotes the quantity representing the size of the matrix, $\mathcal{Q} = \max_i \|q_i\|_2$, $\mathcal{S} = \max_j \|s_j\|_2$, $L_\ell$ and $\mathcal{B}$ are defined in Lemma 3.1, and $C$ is a constant.*

The detailed proof of Lemma 3.2 is provided in the Appendix B. By combining Lemma 3.1 and Lemma 3.2, we can upper bound $\mathcal{L}(\boldsymbol{X})$ of the OCMC model as follows.

**Theorem 3.3.** *For the problem (9), the generalization error $\mathcal{L}(\boldsymbol{X})$ can be upper-bounded by the following expression*

*with probability at least $1 - \delta$,*

$$\mathcal{L}(\boldsymbol{X}) \leq \hat{\mathcal{L}}(\boldsymbol{X}) + \mathcal{B}\sqrt{\frac{\log(1/\delta)}{2|\Omega|}}$$

$$+ 2\min \left\{ 2L_\ell \mathcal{Q}\mathcal{S}\mathcal{P}\sqrt{\frac{\log 2N}{|\Omega|}}, \sqrt{\frac{9CL_\ell \mathcal{X}\mathcal{B}(\sqrt{m} + \sqrt{n})}{|\Omega|}} \right\},$$

*where the related constants are defined in Lemma 3.2.[2]*

Building on Theorem 3.3, the following lemma provides the sample complexity guarantee of our OCMC model.

**Corollary 3.4.** *For any $\epsilon > 0$, we define the $\epsilon$-error recovery of $\boldsymbol{R}$ if $\mathbb{E}_{(i,j)}[\ell(f(i,j), R_{ij})] \leq \epsilon$. Then, when the number of observations is on the order of*

$$\min \left\{ O\left(\frac{\mathcal{P}^2 \log N}{\epsilon^2}\right), O\left(\frac{\mathcal{X}^2 \sqrt{N}}{\epsilon^2}\right) \right\},$$

*an $\epsilon$-error recovery can be achieved by solving (9).*

To further analyze the number of observations according to Corollary 3.4, we can adjust $\lambda_1$ and $\lambda_2$ to set the value of $\mathcal{P}$ as $\|\mathrm{P}_{\boldsymbol{A}^\perp \boldsymbol{B}^\perp}(\boldsymbol{R})\|_*$ or $\|\tilde{\boldsymbol{A}}\boldsymbol{M}^{22}\tilde{\boldsymbol{B}}^T\|_*$ in (4). As shown in the Figure 2, the nuclear norm of the orthogonal complement projection $\|\mathrm{P}_{\boldsymbol{A}^\perp \boldsymbol{B}^\perp}(\boldsymbol{R})\|_*$ decreases approximately linearly with the level of completeness, suggesting that $\mathcal{P}$ follows an approximately linear trend. Consequently, based on the first term in Corollary 3.4, we can infer that as the completeness level increases, the sample complexity decreases at an approximately quadratic rate. This aligns with our intuitive understanding: the greater the completeness of the obtained side information, the more beneficial it is for our matrix completion task. More discussion about the number of observations are provided in Appendix D.

## 4. Linear ADMM Algorithm

To solve the problem (7), we firstly introduce a variable $\boldsymbol{Y}$ as an auxiliary variable, then (7) is reformulated as follows:

$$\min_{\boldsymbol{X}, \boldsymbol{Y}} \quad \|\boldsymbol{X}\|_* + \lambda\|\boldsymbol{Y}\|_*$$

$$\text{s.t. } \mathrm{P}_\Omega(\boldsymbol{X}) = \mathrm{P}_\Omega(\boldsymbol{R}), \quad \boldsymbol{Y} = \mathrm{P}_{\boldsymbol{A}^\perp \boldsymbol{B}^\perp}(\boldsymbol{X}). \quad (13)$$

The augmented Lagrangian function of (13) is given by

$$\mathfrak{L}(\boldsymbol{X}, \boldsymbol{Y}, \boldsymbol{M}_1, \boldsymbol{M}_2, \beta) = \|\boldsymbol{X}\|_* + \lambda\|\boldsymbol{Y}\|_*$$

$$+ \frac{\beta}{2}\|\mathrm{P}_\Omega(\boldsymbol{X}) - \mathrm{P}_\Omega(\boldsymbol{R})\|_F^2 + \frac{\beta}{2}\|\boldsymbol{Y} - \mathrm{P}_{\boldsymbol{A}^\perp \boldsymbol{B}^\perp}(\boldsymbol{X})\|_F^2$$

$$+ \langle \boldsymbol{M}_1, \mathrm{P}_\Omega(\boldsymbol{X}) - \mathrm{P}_\Omega(\boldsymbol{R}) \rangle + \langle \boldsymbol{M}_2, \boldsymbol{Y} - \mathrm{P}_{\boldsymbol{A}^\perp \boldsymbol{B}^\perp}(\boldsymbol{X}) \rangle, \quad (14)$$

where $\boldsymbol{M}_1, \boldsymbol{M}_2$ are the Lagrangian multipliers, $\beta$ is the penalty parameter. and $\langle \cdot, \cdot \rangle$ denotes the matrix inner product: $\langle \boldsymbol{A}, \boldsymbol{B} \rangle = \mathrm{Tr}(\boldsymbol{A}\boldsymbol{B}^T)$. For simplicity, in this paper, we

---

[2]Our analysis makes no assumption on the sampling distribution, making it more general than uniform-sampling-based results (Xu et al., 2013; Recht, 2011). The connection between $\mathcal{L}(\boldsymbol{X})$ and mean squared error is discussed in Appendix C.

focus on the squared loss $\| \cdot \|_F^2$, while the algorithm can be extended to accommodate any loss function.

---

**Algorithm 1** Linearized ADMM for OCMC with Squared Loss

---

1: **Input:** side information matrices $\boldsymbol{A}, \boldsymbol{B}$; sampled matrix $\mathrm{P}_\Omega(\boldsymbol{R})$; maximum iterations $K$; parameters $\lambda, \beta_0, \beta_{\max}, \rho, \tau$
2: **Output:** $\boldsymbol{X}, \boldsymbol{Y}, \boldsymbol{M}_1, \boldsymbol{M}_2$
3: Initialize $\boldsymbol{X}^0 = \boldsymbol{Y}^0 = \boldsymbol{M}_1^0 = \boldsymbol{M}_2^0 = \boldsymbol{0}, k = 0$
4: **while** $k < K$ **do**
5:     Update $\boldsymbol{X}^{k+1}$ by solving (15)
6:     Update $\boldsymbol{Y}^{k+1}$ by solving (16)
7:     Update $\boldsymbol{M}_1^{k+1}$ by (17)
8:     Update $\boldsymbol{M}_2^{k+1}$ by (18)
9:     Update $\beta^{k+1}$ as: $\beta^{k+1} = \min\{\beta_{\max}, \rho\beta^k\}$
10:    $k = k + 1$
11: **end while**

---

To solve the problem (14) iteratively, let $k$ denote the iteration index. Given the current values $\boldsymbol{X}^k, \boldsymbol{Y}^k, \boldsymbol{M}_1^k, \boldsymbol{M}_2^k$, the variables can be updated as follows:

$$\boldsymbol{X}^{k+1} = \arg\min_{\boldsymbol{X}} \mathfrak{L}(\boldsymbol{X}, \boldsymbol{Y}^k, \boldsymbol{M}_1^k, \boldsymbol{M}_2^k, \beta), \tag{15}$$

$$\boldsymbol{Y}^{k+1} = \arg\min_{\boldsymbol{Y}} \mathfrak{L}(\boldsymbol{X}^{k+1}, \boldsymbol{Y}, \boldsymbol{M}_2^k, \beta), \tag{16}$$

$$\boldsymbol{M}_1^{k+1} = \boldsymbol{M}_1^k + \beta(\mathrm{P}_\Omega(\boldsymbol{X}^{k+1}) - \mathrm{P}_\Omega(\boldsymbol{R})), \tag{17}$$

$$\boldsymbol{M}_2^{k+1} = \boldsymbol{M}_2^k + \beta(\boldsymbol{Y}^{k+1} - \mathrm{P}_{\boldsymbol{A}^\perp \boldsymbol{B}^\perp}(\boldsymbol{X}^{k+1})). \tag{18}$$

the derivations of the closed solution of (15) to (18) and complexity analysis are provided in Appendix E. The workflow of our algorithm is summarized in Algorithm 1.

This enhances computational efficiency for each subproblem can be solved iteratively and in parallel. Moreover, since the problem in (7) is convex, Algorithm 1 will converge to a global optimal solution after a finite number of iterations (Yang & Yuan, 2013) and the convergence rate of linear ADMM is $O(1/k)$ (Fang et al., 2015; Shi et al., 2015).

## 5. Experimental Results

In this section, we demonstrate the effectiveness of the proposed OCMC model and the linear ADMM algorithm through experiments conducted on both synthetic experiments and real-world applications. In the synthetic experiments, we primarily investigated the relationship between matrix completion accuracy and the number of observations under varying completeness levels of side information. In the real-world application experiments, we focused on the multi-label learning (Goldberg et al., 2010) and movie recommendation (Harper & Konstan, 2015).

### 5.1. Synthetic Data Experiments

**Settings and Baseline:** We set $\boldsymbol{J}, \boldsymbol{K} \in \mathbb{R}^{100 \times 10}$ with elements $J_{ij}, K_{ij} \sim \mathcal{N}(0, 1)$, generating the low rank matrix

$\boldsymbol{R} = \boldsymbol{J}\boldsymbol{K}^T \in \mathbb{R}^{100 \times 100}$ of rank 10. Side information matrices $\boldsymbol{A}$ and $\boldsymbol{B}$ are derived from the SVD of $\boldsymbol{R} = \boldsymbol{U}\boldsymbol{\Sigma}\boldsymbol{V}^T$, where $\boldsymbol{U}, \boldsymbol{V} \in \mathbb{R}^{100 \times 10}$ and $\boldsymbol{\Sigma} \in \mathbb{R}^{10 \times 10}$. We then transform $\boldsymbol{U}$ and $\boldsymbol{V}$ by multiplying them with random matrices. Specifically, $\boldsymbol{U}$ is transformed as $\boldsymbol{A} = \boldsymbol{U}\boldsymbol{T}$, where $\boldsymbol{T} \in \mathbb{R}^{10 \times d}$, with $T_{ij} \sim \mathcal{N}(0, 1)$, and $\boldsymbol{B} = \boldsymbol{V}\boldsymbol{Q}$, following the similar transformation, where $\boldsymbol{Q}$ is a random matrix generated similarly to $\boldsymbol{T}$. Making $\boldsymbol{T}$ and $\boldsymbol{Q}$ column full rank, the integer $d$ is set as $d = r \times \eta$, where $\eta$ is the completeness level. For example, with 50% completeness, $\eta = 50\%$, so $d = 10 \times 50\% = 5$. For simplicity, in our experiments, when we refer to the completeness of side information as 50%, it means that the completeness of both the row and column side information is 50%.

Three representative algorithms are selected as baselines: SVT(Cai et al., 2010), Maxide(Xu et al., 2013), and Dirty-IMC(Chiang et al., 2015). We use the relative Frobenius norm difference as the performance metric: error $= \|\boldsymbol{R} - \hat{\boldsymbol{R}}\|_F / \|\boldsymbol{R}\|_F$, where $\boldsymbol{R}$ is the target matrix and $\hat{\boldsymbol{R}}$ is the completed matrix. The sampling rate is defined as $|\Omega|/m \times n$, where $\Omega$ is the set of the indices of observed entries, and $m, n$ represent the target matrix dimension.

In Figure 4, we evaluate the completion accuracy of the proposed OCMC under different sampling rates. To ensure a comprehensive evaluation, we consider three levels of completeness: low ($\eta = 20\%$), medium ($\eta = 50\%$), and high ($\eta = 80\%$). More experiments are provided in Appendix F.

**Results:** From Figure 4, we can observe that our proposed OCMC consistently outperforms the established baselines across all three levels of completeness. With the exception of the Maxide, the completion accuracy of all matrix completion models improves as the sampling rate increases. The completion accuracy of the Maxide remains at a relatively low level. We conjecture that it is because in incomplete situation, Maxide restricts the feasible set based on incorrect row and column information, excluding the true matrix from this feasible set. This situation further emphasizes the importance of methods for using incomplete side information. Among the remaining algorithms, SVT exhibits the lowest accuracy, because it does not use any side information.

### 5.2. Multi-label Learning

In multi-label learning, each instance can belong to multiple categories simultaneously. Since labels and features are often interrelated, and jointly low rank, matrix completion methods have been introduced to capture the latent correlations (Goldberg et al., 2010).

**Settings and Baseline:** We compared our proposed OCMC model with the SVT, Maxide, DirtyIMC, as well as FPC (Ma et al., 2011a) and FNNM(Yang et al., 2020), which shows promising results in various practical scenarios. The dataset

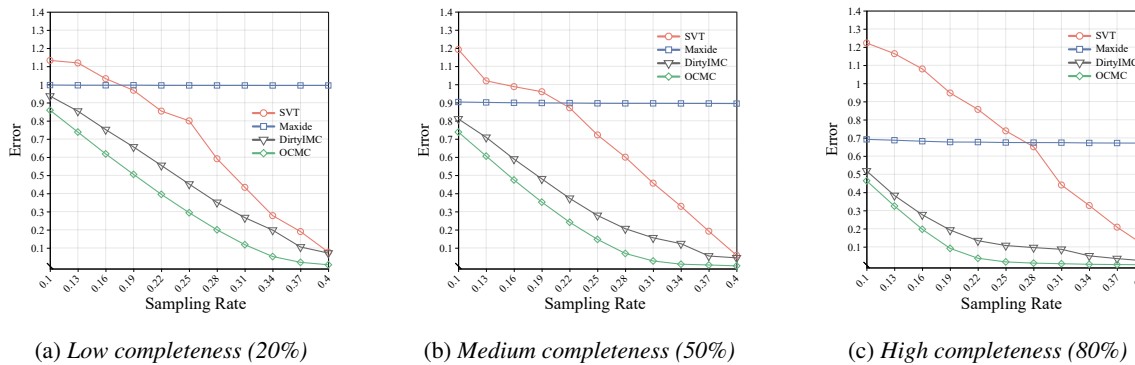

(a) Low completeness (20%)      (b) Medium completeness (50%)      (c) High completeness (80%)

Figure 4. Error VS. Sampling rate under different completion rate of side information.

Table 2. Comparison of algorithms on the "Arts" dataset with varying sampling rates $\omega$.

| DATASET | $\omega\%$ | SVT | MAXIDE | DIRTYIMC | FPC | OCMC | FNNM |
|---------|-----------|-----|--------|----------|-----|------|------|
| ARTS | 10% | 0.3500 | **0.5746** | 0.4591 | 0.3402 | 0.5249 | 0.5126 |
| | 30% | 0.4639 | 0.6435 | 0.6416 | 0.5012 | **0.7206** | 0.6825 |
| | 50% | 0.6053 | 0.6784 | 0.7648 | 0.6103 | **0.8358** | 0.7625 |
| | 70% | 0.7271 | 0.6991 | 0.8584 | 0.7592 | **0.9029** | 0.8326 |
| | 90% | 0.8695 | 0.7135 | 0.9266 | 0.8763 | **0.9596** | 0.9115 |

we selected is the web page classification from "Yahoo.com" (Ueda & Saito, 2002), which includes 11 distinct topics. For instance, in the "Art" category, there are 5,000 instances, each with 432 features and 21 label categories.

We randomly pick 10% instances as the test set and use the rest 90% as the training set. To conduct partial label assignment in the training set, for each label, we randomly choose 10% positive and negative training instances and keep the remaining training instances unknown. The percentage of training instances $\omega\%$ ranges from 10% to 90% with an increasing step size of 20%. Notably, $\omega\%$ also represents the sampling rate. In this experiment, we use Average Precision (AP) (Mazumder et al., 2010) as the evaluation metric.

**Results:** Table 2 summarizes the results for the "Arts" category. The best AP result is **bold** and second best result is underline. It can be learned that our OCMC achieves four best results and one second best result. These results demonstrate the effectiveness of the proposed OCMC model in leveraging incomplete side information to improve performance in multi-label learning applications. Comparisons for other categories are provided in Appendix F for reference.

### 5.3. MovieLens-100k

In this experiment, we use the MovieLens-100k dataset (Harper & Konstan, 2015), which contains 100,000 ratings (ranging from 1 to 5) from 943 users on 1,682 movies, and includes 23 user features (e.g., age, gender, occupation) and 20 movie features (e.g., genre, release date).

**Settings and Baseline:** The data is divided into a training set and a test set, with the proportion of the test set ranging from 0.1 to 0.9 in increments of 0.2. The performance of different methods is evaluated by the Root Mean Square Error (RMSE) on the test set, defined as:

$$\text{RMSE} = \frac{1}{|\tilde{\Omega}|}\sqrt{\sum_{(i,j)\in\tilde{\Omega}}(X_{ij} - R_{ij})^2},$$

where $R_{ij}$ is the sampled rating, $X_{ij}$ is the corresponding completed rating, and $\tilde{\Omega}$ represents the indices in the test set. Like in multi-label learning, we choose SVT, Maxide, DirtyIMC, FPC and FNNC as baselines.

**Results:** The RMSE values for all methods at different sampling rates are shown in Table 3. The best AP result

Table 3. RMSE of algorithms on the "MovieLens-100k" dataset with the varying training set proportion.

| PROPORTION | SVT | MAXIDE | DIRTYIMC | OCMC | FPC | FNNM |
|-----------|-----|--------|----------|------|-----|------|
| 10% | 2.3824 | 1.4512 | 1.3053 | **1.1503** | 1.8050 | 1.3533 |
| 30% | 1.5523 | 1.4483 | 1.0251 | **0.9826** | 1.2249 | 1.0466 |
| 50% | 1.3325 | 1.4415 | 0.9532 | **0.9013** | 1.0733 | 0.9701 |
| 70% | 1.2284 | 1.4392 | 0.9321 | **0.8872** | 1.0033 | 0.9283 |
| 90% | 1.1423 | 1.4336 | 0.8702 | **0.8625** | 0.9533 | 0.8983 |

is **bold** and second best result is underline. As the proportion increases, the performance of most methods improves. Among them, DirtyIMC, FNNM and OCMC are applicable for matrix completion with incomplete side information, demonstrating better completion performance compared to other methods. Notably, our proposed OCMC model achieves the best performance among all benchmarks, demonstrating its advantage in effectively leveraging incomplete side information for accurate recommendations.

### 5.4. Discussion

The experimental results on real-world datasets, as demonstrated in the previous subsections, show that OCMC effectively captures the characteristics of practical side information. This can be attributed to the fact that the proposed OCMC is designed for scenarios where side information is available but incomplete, which is a common situation in real-world applications. In the following, we justify why side information is often incomplete in practice, from both intuitive and geometric perspectives.

- *Intuitive and practical perspective:* In recommendation systems such as MovieLens-100k, side information typically includes attributes of users (e.g., age, gender) and items (e.g., categories, genres). However, actual user preferences and item characteristics are influenced by additional latent factors that are not directly available but still affect the rating matrix. Thus, it is unrealistic to assume that a few available features fully describe user behavior or item semantics. A similar situation occurs in multi-label learning, where side information such as feature descriptors or annotations only partially reflects the complex dependencies among labels.

- *Subspace-based geometric perspective:* If the side information were complete, the column (or row) space of the target matrix would lie entirely within the subspace spanned by the given side information. However, this assumption does not hold in the datasets we studied. A formal check involves projecting the target matrix onto the side information subspace and evaluating whether the projection error is zero. Specifically, we can check whether the residual norm $\|\boldsymbol{P}_A\boldsymbol{R} - \boldsymbol{R}\|_F = 0$, where $\boldsymbol{P}_A = \boldsymbol{A}(\boldsymbol{A}^\top\boldsymbol{A})^{-1}\boldsymbol{A}^\top$. A large residual norm indicates that the given side information does not fully span the target subspace. To illustrate this, the residual norms of the experimental datasets are shown in Table 4. As observed, these residual norms are large compared to the norms of the target matrix, confirming the incompleteness of the side information.

These insights confirm that side information in real-world datasets is generally incomplete. This also helps explain the strong empirical performance of OCMC on benchmarks such as recommendation systems and multi-label learning.

*Table 4.* We compute the residual norm: $\|\boldsymbol{P}_A\boldsymbol{R} - \boldsymbol{R}\|_F$ and its normalized form: $\|\boldsymbol{P}_A\boldsymbol{R} - \boldsymbol{R}\|_F/\|\boldsymbol{R}\|_F$ of each dataset from MovieLens-100K and 11 datasets of "Yahoo.com".

| CATEGORY | RESIDUAL NORM | NORMALIZED |
|---|---|---|
| MOVIELENS-100K | 1122.5 | 0.9581 |
| ARTS | 71.2843 | 0.7882 |
| BUSINESS | 53.3929 | 0.5992 |
| COMPUTERS | 61.7102 | 0.7111 |
| ENTERTAINMENT | 60.3578 | 0.7162 |
| EDUCATION | 65.2612 | 0.7637 |
| HEALTH | 60.2478 | 0.6609 |
| RECREATION | 64.6424 | 0.7663 |
| REFERENCE | 52.4684 | 0.6862 |
| SCIENCE | 65.0822 | 0.7642 |
| SOCIAL | 49.5415 | 0.6184 |
| SOCIETY | 69.95 | 0.7605 |

## 6. Conclusion

In this paper, we investigated matrix completion problem with incomplete side information. A novel OCMC model is proposed that leverages the low-rank properties of both the target matrix and its orthogonal complement projection. Using the PAC framework, we theoretically establish a generalization error bound for the OCMC model and show that the sample complexity decreases quadratically as the completeness level increases. Additionally, a linear ADMM algorithm is proposed to efficiently solve the OCMC model, which is guaranteed to converge to a global optimal solution. Experiments show that the OCMC model outperforms the existing matrix completion models in both synthetic and real-world applications, including multi-label learning and movie recommendation. As future work, we plan to integrate the proposed OCMC framework into neural network-based approaches to more effectively exploit side information and improve performance.

### Acknowledgements

This work was supported in part by the National Natural Science Foundation of China under Grant No. 62401180, the Shenzhen Science and Technology R&D Funds under Grant No. GXWD20231130104830003, the Guangdong Basic and Applied Basic Research Foundation under Grant No. 2025A1515011733.

### Impact Statement

This paper presents work whose goal is to advance the field of Machine Learning. There are many potential societal consequences of our work, none which we feel must be specifically highlighted here.

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

Here, we present a detailed description of related work in Appendix A. Detailed proofs of Lemma 3.2 and Corollary 3.4 are provided in Appendix B and Appendix D respectively, which are the two most crucial parts of our theory. Appendix E provides a detailed derivation of Algorithm 1. More experiments are presented in Appendix F and the full table of multilabel learning outcome is also in the Appendix F.

## A. Related Work

**Transductive Matrix Completion:** In most cases, matrix completion refers to *transductive matrix completion*. This approach focuses on completing a low-rank matrix using a limited number of sampled entries, relying only on the low-rank property of the matrix. Early studies on matrix completion were developed for applications like collaborative filtering , such as the famous Netflix Prize (Koren, 2009; Töscher et al., 2009). Several theoretical foundations have been established in this field.

One remarkable milestone was established in (Candes & Recht, 2012; Recht, 2011)., which demonstrated that, if the entries of the desired matrix are uniformly sampled, solving the nuclear norm model

$$\min_{\boldsymbol{X}} \|\boldsymbol{X}\|_* \quad \text{s.t. } \mathrm{P}_\Omega(\boldsymbol{X}) = \mathrm{P}_\Omega(\boldsymbol{R})$$

allows one to perfectly complete a low-rank matrix $\boldsymbol{X} \in \mathbb{R}^{m \times n}$ from $O(N \log^2 N)$ samples, where $N = \max\{m, n\}$. (Recht, 2011) demonstrates the theory from the perspective of sampling with replacement, arriving at the same sample complexity results. Based on these basis, a series of convex relaxation algorithms have been developed. As mentioned above, (Candes & Recht, 2012) proposed a nuclear norm relaxation framework, while (Cai et al., 2010) introduced the singular value thresholding approach. By calculating the following iteration, SVT combines nuclear norm relaxation with proximal gradient methods:

$$\begin{cases} \boldsymbol{X}_k = \mathrm{D}_\tau(\boldsymbol{Y}_{k-1}) \\ \boldsymbol{Y}_k = \boldsymbol{Y}_{k-1} + \delta_k \mathrm{P}_\Omega(\boldsymbol{R} - \boldsymbol{X}_k). \end{cases}$$

Some studies also employ variants of the nuclear norm to approximate the rank of a matrix, such as (Gu et al., 2017; Nie et al., 2012).

Another popular approach is by *matrix factorization*, which has become a widely applicable and empirically successful method for low-rank matrix completion. In this kind of approaches, the low-rank target matrix is represented in a bilinear form:

$$\boldsymbol{X} = \boldsymbol{U}\boldsymbol{V}^T,$$

where $\boldsymbol{U} \in \mathbb{R}^{m \times p}, \boldsymbol{V} \in \mathbb{R}^{n \times p}$. By constraining the dimensions of the factor matrices, i.e., the size of $p$, we can effectively limit the rank of the target matrix based on the principles of matrix multiplication.

The most known algorithm through matrix factorization is the *alternating minimization* (Jain et al., 2013), which formed a major component of the winning entry in the Netflix Challenge.This algorithm performs matrix completion by solving the following problem:

$$\min_{\boldsymbol{U} \in \mathbb{R}^{m \times p}, \ \boldsymbol{V} \in \mathbb{R}^{n \times p}} \|\mathrm{P}_\Omega(\boldsymbol{U}\boldsymbol{V}^\top) - \mathrm{P}_\Omega(\boldsymbol{R})\|_F^2 \to \begin{cases} \min_{\boldsymbol{U} \in \mathbb{R}^{m \times p}} \|\mathrm{P}_\Omega(\boldsymbol{U}\boldsymbol{V}^{k^\top}) - \mathrm{P}_\Omega(\boldsymbol{R})\|_F^2 \\ \min_{\boldsymbol{V} \in \mathbb{R}^{n \times p}} \|\mathrm{P}_\Omega(\boldsymbol{U}^k\boldsymbol{V}^\top) - \mathrm{P}_\Omega(\boldsymbol{R})\|_F^2 \end{cases}.$$

Although the overall problem is non-convex, each sub-problem is typically convex and can be solved efficiently. (Jain et al., 2013) provided convergence guarantees for alternating minimization in matrix completion. In addition, several gradient-based algorithms have proven effective. For instance, (Raghunandan H. Keshavan, 2010) proposed the *Spectral Matrix Completion method*, considering a variant of matrix completion.

**Inductive Matrix Completion:** Inductive matrix completion leverages side information about the row and column spaces of the matrix to improve the accuracy of the completion process. Various algorithms have been applied across domains like collaborative filtering , response prediction, and gene-disease association prediction , significantly enhancing efficiency. Most of these algorithms rely on various matrix factorization techniques.

For example, studies by (Jain & Dhillon, 2013), as well as (Xu et al., 2013), focused on matrix factorization for the matrix

completion problem with side information. Both approaches utilize a representation trick:

$$\boldsymbol{R} = \boldsymbol{AMB}^T,$$

where $\boldsymbol{R}$ is the desired matrix, and $\boldsymbol{A}$ and $\boldsymbol{B}$ are side information matrices corresponding to the row and column spaces, respectively. Their results suggest that, by optimizing

$$\min_{\boldsymbol{M}} \|\boldsymbol{M}\|_* \quad \text{s.t. } \mathrm{P}_\Omega(\boldsymbol{AMB}) = \mathrm{P}_\Omega(\boldsymbol{R}),$$

where the sample complexity can be reduced to $O(\log N)$.

Considering the latent noise in side information, a more recent study (Chiang et al., 2015) generalized this model by introducing a *dirty statistic* model, given by:

$$\boldsymbol{X} = \boldsymbol{AMB}^T + \boldsymbol{N}.$$

Then, they optimize the following problem:

$$\min_{\boldsymbol{M,N}} \|\boldsymbol{M}\|_* + \|\boldsymbol{N}\|_* \quad \text{s.t. } \mathrm{P}_\Omega(\boldsymbol{AMB}^T + \boldsymbol{N}) = \mathrm{P}_\Omega(\boldsymbol{R}).$$

This work proposes a method for evaluating the quality of side information and demonstrates that sample complexity can be reduced even with noisy side information, depending on its quality.

Additionally, (Lu et al., 2016) focused on the embedded matrix $\boldsymbol{M}$ and proposed a sparse interaction model:

$$\min_{\boldsymbol{M,E}} \frac{1}{2}\|\boldsymbol{XMY}^T - \boldsymbol{E}\|_F^2 + \lambda_M\|\boldsymbol{M}\|_1 + \lambda_E\|\boldsymbol{E}\|_*, \quad \text{subject to } \mathrm{P}_\Omega(\boldsymbol{E}) = \mathrm{P}_\Omega(\boldsymbol{R}).$$

which achieves similar results on the effectiveness of imperfect side information in matrix completion.

## B. Proof of Lemma 3.2

*Proof.* We separately prove that the Rademacher complexity is less than each term on the right side in Lemma 3.2. Specifically, we will show that

$$\mathfrak{R}_\ell(\mathcal{F}) \leq 2L_\ell \mathcal{Q} \mathcal{S} \mathcal{P} \sqrt{\frac{\log 2N}{|\Omega|}}, \tag{19}$$

$$\mathfrak{R}_\ell(\mathcal{F}) \leq \sqrt{\frac{9CL_\ell \mathcal{X} \mathcal{B}(\sqrt{m} + \sqrt{n})}{|\Omega|}}. \tag{20}$$

To prove the first inequality in (19), we mainly use the following lemma.

**Lemma B.1.** *(Kakade et al., 2008) Let $S_x = \{\boldsymbol{W} \in \mathbb{R}^{n \times n} \mid \|\boldsymbol{W}\|_* \leq \mathcal{W}\}$, $\mathcal{A} = \max_{i \in \{1,2,...,m\}} \|\boldsymbol{A}_i\|_2$, $\boldsymbol{A}_i \in \mathbb{R}^{n \times n}$, we have*

$$\mathbb{E}_\sigma \left[ \sup_{\boldsymbol{W} \in S_x} \frac{1}{m} \sum_{i=1}^m \sigma_i \, \mathrm{trace}(\boldsymbol{W}\boldsymbol{A}_i) \right] \leq 2\mathcal{A}\mathcal{W}\sqrt{\frac{\log 2n}{m}}.$$

This Lemma is a corollary of Theorem 1 in (Kakade et al., 2008).

Recalling the definition of the Rademacher complexity, the Rademacher complexity of $\mathcal{F}$ is given by

$$\mathfrak{R}_\ell(\mathcal{F}) = \mathbb{E}_\sigma \left[ \sup_{\substack{\|\boldsymbol{X}\|_* \leq \mathcal{X} \\ \|\mathrm{P}_{\boldsymbol{A}^\perp \boldsymbol{B}^\perp}(\boldsymbol{X})\|_* \leq \mathcal{P}}} \frac{1}{|\Omega|} \sum_{\alpha=1}^{|\Omega|} \sigma_\alpha \ell(X_{i_\alpha j_\alpha}, R_{i_\alpha j_\alpha}) \right]. \tag{21}$$

Using the contraction principle of Rademacher complexity(see (Bartlett & Mendelson, 2002)), the expression in (21) is bounded by

$$\mathfrak{R}_\ell(\mathcal{F}) \leq \mathbb{E}_\sigma \left[ \sup_{\substack{\|\boldsymbol{X}\|_* \leq \mathcal{X} \\ \|\mathrm{P}_{\boldsymbol{A}^\perp \boldsymbol{B}^\perp}(\boldsymbol{X})\|_* \leq \mathcal{P}}} \frac{L_\ell}{|\Omega|} \sum_{\alpha=1}^{|\Omega|} \sigma_\alpha X_{i_\alpha j_\alpha} \right]. \tag{22}$$

We define the matrix $\boldsymbol{\Gamma}$ as follows:

$$\Gamma_{ij} = \begin{cases} \sigma_\alpha, & \text{if } i = i_\alpha \text{ and } j = j_\alpha \\ 0, & \text{otherwise.} \end{cases}$$

Then, by representing the inner product of matrices, we can rewrite (22) as follows:

$$\mathfrak{R}_\ell(\mathcal{F}) \leq \mathbb{E}_\sigma \left[ \sup_{\substack{\|\boldsymbol{X}\|_* \leq \mathcal{X} \\ \|\mathrm{P}_{\boldsymbol{A}^\perp \boldsymbol{B}^\perp}(\boldsymbol{X})\|_* \leq \mathcal{P}}} \frac{L_\ell}{|\Omega|} \langle \boldsymbol{\Gamma}, \boldsymbol{X} \rangle \right] = \mathbb{E}_\sigma \left[ \sup_{\substack{\|\boldsymbol{X}\|_* \leq \mathcal{X} \\ \|\mathrm{P}_{\boldsymbol{A}^\perp \boldsymbol{B}^\perp}(\boldsymbol{X})\|_* \leq \mathcal{P}}} \frac{L_\ell}{|\Omega|} \, \mathrm{tr}(\boldsymbol{X}^T \boldsymbol{\Gamma}) \right]. \tag{23}$$

To proceed with (23), we let $\boldsymbol{Q} \in \mathbb{R}^{m \times m}$ be a matrix with columns $\boldsymbol{Q} = \{q_i\}$, and let $\boldsymbol{S} \in \mathbb{R}^{n \times n}$ be a matrix with columns $\boldsymbol{S} = \{s_i\}$, where $(\boldsymbol{I} + \mu \boldsymbol{I} - \boldsymbol{P_B})\boldsymbol{S} = \boldsymbol{I} = \boldsymbol{Q}(\boldsymbol{I} + \mu \boldsymbol{I} - \boldsymbol{P_A})$ for some $\mu > 0$. Then, we have

$$\mathrm{tr}(\boldsymbol{X}^T \boldsymbol{\Gamma}) = \mathrm{tr}\left[\boldsymbol{Q}(\boldsymbol{I} + \mu \boldsymbol{I} - \boldsymbol{P_A})\boldsymbol{X}(\boldsymbol{I} + \mu \boldsymbol{I} - \boldsymbol{P_B})\boldsymbol{S}\boldsymbol{\Gamma}^T\right].$$

Then, by substituting $\mathrm{tr}(\boldsymbol{X}^T \boldsymbol{\Gamma})$ in (23), we have the following inequality:

$$\mathfrak{R}_\ell(\mathcal{F}) \leq \mathbb{E}_\sigma \left\{ \sup_{\substack{\|\boldsymbol{X}\|_* \leq \mathcal{X} \\ \|\mathrm{P}_{\boldsymbol{A}^\perp \boldsymbol{B}^\perp}(\boldsymbol{X})\|_* \leq \mathcal{P}}} \frac{L_\ell}{|\Omega|} \, \mathrm{tr}\left[\boldsymbol{Q}(\boldsymbol{I} + \mu \boldsymbol{I} - \boldsymbol{P_A})\boldsymbol{X}(\boldsymbol{I} + \mu \boldsymbol{I} - \boldsymbol{P_B})\boldsymbol{S}\boldsymbol{\Gamma}^T\right] \right\}. \tag{24}$$

To apply the Lemma B.1 for (24), we consider the following definition.

**Definition B.2** ($\mu$-relaxation for operation $\mathrm{P}_{\boldsymbol{A}^\perp \boldsymbol{B}^\perp}(\cdot)$). *For any matrix $\boldsymbol{R} \in \mathbb{R}^{m \times n}$ and $\mu > 0$, we define the linear operation $\mathrm{P}_{\boldsymbol{A}^\perp \boldsymbol{B}^\perp}^\mu(\cdot)$ as the $\mu$-relaxation for $\mathrm{P}_{\boldsymbol{A}^\perp \boldsymbol{B}^\perp}(\cdot)$:*

$$\mathrm{P}_{\boldsymbol{A}^\perp \boldsymbol{B}^\perp}^\mu(\boldsymbol{R}) = (\boldsymbol{I} + \mu \boldsymbol{I} - \boldsymbol{P_A})\boldsymbol{R}(\boldsymbol{I} + \mu \boldsymbol{I} - \boldsymbol{P_B}).$$

*The operation $\mathrm{P}_{\boldsymbol{A}^\perp \boldsymbol{B}^\perp}(\cdot)$ can be viewed as $\lim_{\mu \to 0} \mathrm{P}_{\boldsymbol{A}^\perp \boldsymbol{B}^\perp}^\mu(\cdot)$. Here, $\mu$ is called the relaxation factor.*

The advantage of Definition B.2 is that it allows the projection operation to become an invertible operation while maintaining the constraints to some extent, as $\mu$ can be arbitrarily small. This facilitates analysis and the generalization of the results. Based on Definition B.2, we can intuitively see that when $\mu$ is very small, the constraint on $\|\mathrm{P}_{\boldsymbol{A}^\perp \boldsymbol{B}^\perp}^\mu(\boldsymbol{X})\|_*$ should yield a result very close to $\|\mathrm{P}_{\boldsymbol{A}^\perp \boldsymbol{B}^\perp}(\boldsymbol{X})\|_*$. The simulation results in Figure 5 are also consistent with this observation. Thus, we can approximate the constraint $\|\mathrm{P}_{\boldsymbol{A}^\perp \boldsymbol{B}^\perp}(\boldsymbol{X})\|_* \leq \mathcal{P}$ by $\|\mathrm{P}_{\boldsymbol{A}^\perp \boldsymbol{B}^\perp}^\mu(\boldsymbol{X})\|_* \leq \mathcal{P}$ in (24).

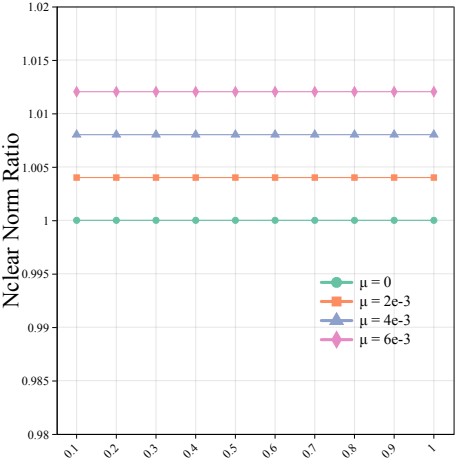

*Figure 5.* Nuclear norm ratio $\|\mathrm{P}_{\boldsymbol{A}^\perp \boldsymbol{B}^\perp}^\mu(\boldsymbol{X})\|_* / \|\mathrm{P}_{\boldsymbol{A}^\perp \boldsymbol{B}^\perp}(\boldsymbol{X})\|_*$ *for $\mu \in \{0, 2 \times 10^{-3}, 4 \times 10^{-3}, 6 \times 10^{-3}\}$ and completeness levels in $\{0.1, \cdots, 1\}$, where $\boldsymbol{X}$ is a randomly generated $100 \times 100$ matrix. The experiments were repeated ten times, and the average results are shown. The maximum difference in nuclear norms, at the same completeness level, is less than 1.5% across different values of $\mu$.*

Therefore, we can express (24) as follows

$$\mathfrak{R}_\ell(\mathcal{F}) \le \mathbb{E}_\sigma \left\{ \sup_{\substack{\|\boldsymbol{X}\|_* \le \mathcal{X} \\ \|\mathrm{P}^\mu_{\boldsymbol{A}^\perp \boldsymbol{B}^\perp}(\boldsymbol{X})\|_* \le \mathcal{P}}} \frac{L_\ell}{|\Omega|} \sum_{\alpha=1}^{|\Omega|} \sigma_\alpha \left[ \boldsymbol{Q}(\boldsymbol{I} + \mu\boldsymbol{I} - \boldsymbol{P_A})\boldsymbol{X}(\boldsymbol{I} + \mu\boldsymbol{I} - \boldsymbol{P_B})\boldsymbol{S} \right]_{i_\alpha j_\alpha} \right\}$$

$$= \mathbb{E}_\sigma \left\{ \sup_{\substack{\|\boldsymbol{X}\|_* \le \mathcal{X} \\ \|\mathrm{P}^\mu_{\boldsymbol{A}^\perp \boldsymbol{B}^\perp}(\boldsymbol{X})\|_* \le \mathcal{P}}} \frac{L_\ell}{|\Omega|} \sum_{\alpha=1}^{|\Omega|} \sigma_\alpha \operatorname{tr}\left[ q_{i_\alpha}^T (\boldsymbol{I} + \mu\boldsymbol{I} - \boldsymbol{P_A})\boldsymbol{X}(\boldsymbol{I} + \mu\boldsymbol{I} - \boldsymbol{P_B})s_{j_\alpha} \right] \right\}$$

$$= \mathbb{E}_\sigma \left\{ \sup_{\substack{\|\boldsymbol{X}\|_* \le \mathcal{X} \\ \|\mathrm{P}^\mu_{\boldsymbol{A}^\perp \boldsymbol{B}^\perp}(\boldsymbol{X})\|_* \le \mathcal{P}}} \frac{L_\ell}{|\Omega|} \sum_{\alpha=1}^{|\Omega|} \sigma_\alpha \operatorname{tr}\left[ (\boldsymbol{I} + \mu\boldsymbol{I} - \boldsymbol{P_A})\boldsymbol{X}(\boldsymbol{I} + \mu\boldsymbol{I} - \boldsymbol{P_B})s_{j_\alpha} q_{i_\alpha}^T \right] \right\}$$

$$= \mathbb{E}_\sigma \left\{ \sup_{\substack{\|\boldsymbol{X}\|_* \le \mathcal{X} \\ \|\mathrm{P}^\mu_{\boldsymbol{A}^\perp \boldsymbol{B}^\perp}(\boldsymbol{X})\|_* \le \mathcal{P}}} \frac{L_\ell}{|\Omega|} \sum_{\alpha=1}^{|\Omega|} \sigma_\alpha \operatorname{tr}\left[ \mathrm{P}^\mu_{\boldsymbol{A}^\perp \boldsymbol{B}^\perp}(\boldsymbol{X}) s_{j_\alpha} q_{i_\alpha}^T \right] \right\}, \tag{25}$$

where the final equality uses the definition of $\mathrm{P}^\mu_{\boldsymbol{A}^\perp \boldsymbol{B}^\perp}(\cdot)$ in Definition B.2.

Using Lemma B.1, we let $\mathcal{Q} = \max_i \|q_i\|_2$, $\mathcal{S} = \max_j \|s_j\|_2$, then the expression in (25) can be further bounded by

$$\mathfrak{R}_\ell(\mathcal{F}) \le \mathbb{E}_\sigma \left\{ \sup_{\substack{\|\boldsymbol{X}\|_* \le \mathcal{X} \\ \|\mathrm{P}^\mu_{\boldsymbol{A}^\perp \boldsymbol{B}^\perp}(\boldsymbol{X})\|_* \le \mathcal{P}}} \frac{L_\ell}{|\Omega|} \sum_{\alpha=1}^{|\Omega|} \sigma_\alpha \operatorname{tr}\left[ \mathrm{P}^\mu_{\boldsymbol{A}^\perp \boldsymbol{B}^\perp}(\boldsymbol{X}) s_{j_\alpha} q_{i_\alpha}^T \right] \right\} \le 2L_\ell \mathcal{Q}\mathcal{S}\mathcal{P} \sqrt{\frac{\log 2N}{|\Omega|}}, \tag{26}$$

which concludes the proof of the first lower bound (19) in Lemma 3.2.

For the second lower bound (20), using the same trick in (Lu et al., 2016), we can divide the sampling matrix $\boldsymbol{\Gamma}$ into two parts based on "hit-time." Specifically, by setting a threshold $t$, any entry $\Gamma_{ij}$ exceeding $t$ indicates a frequently sampled element, while values below $t$ suggest infrequent samples. Using this criterion, we obtain matrices $E$ and $F$ as follows:

$$E_{ij} = \begin{cases} \Gamma_{ij}, & \Gamma_{ij} \ge t \\ 0, & \text{otherwise} \end{cases} \qquad F_{ij} = \begin{cases} \Gamma_{ij}, & \Gamma_{ij} < t \\ 0, & \text{otherwise.} \end{cases}$$

Thus, we have $\boldsymbol{\Gamma} = \boldsymbol{E} + \boldsymbol{F}$, and the Rademacher complexity in (23) can be bounded as follows:

$$\mathfrak{R}_\ell(\mathcal{F}) = \mathbb{E}_\sigma \left[ \sup_{\substack{\|\boldsymbol{X}\|_* \le \mathcal{X} \\ \|\mathrm{P}_{\boldsymbol{A}^\perp \boldsymbol{B}^\perp}(\bar{\boldsymbol{X}})\|_* \le \mathcal{P}}} \frac{1}{|\Omega|} \sum_{i,j} \Gamma_{ij} \ell(X_{ij}, R_{ij}) \right]$$

$$= \mathbb{E}_\sigma \left[ \sup_{\substack{\|\boldsymbol{X}\|_* \le \mathcal{X} \\ \|\mathrm{P}_{\boldsymbol{A}^\perp \boldsymbol{B}^\perp}(\bar{\boldsymbol{X}})\|_* \le \mathcal{P}}} \frac{1}{|\Omega|} \sum_{i,j} E_{ij} \ell(X_{ij}, R_{ij}) \right] + \mathbb{E}_\sigma \left[ \sup_{\substack{\|\boldsymbol{X}\|_* \le \mathcal{X} \\ \|\mathrm{P}_{\boldsymbol{A}^\perp \boldsymbol{B}^\perp}(\bar{\boldsymbol{X}})\|_* \le \mathcal{P}}} \frac{1}{|\Omega|} \sum_{i,j} F_{ij} \ell(X_{ij}, R_{ij}) \right]. \tag{27}$$

According to the Definition B.2, we have $\ell(X_{ij}, R_{i_\alpha j_\alpha}) \le \mathcal{B}$, then the first term in (27) can be bounded as follows:

$$\mathbb{E}_\sigma \left[ \sup_{\substack{\|\boldsymbol{X}\|_* \le \mathcal{X} \\ \|\mathrm{P}_{\boldsymbol{A}^\perp \boldsymbol{B}^\perp}(\bar{\boldsymbol{X}})\|_* \le \mathcal{P}}} \frac{1}{|\Omega|} \sum_{ij} E_{ij} \ell(X_{ij}, R_{i_\alpha j_\alpha}) \right] \le \mathbb{E}_\sigma \left[ \sup_{\substack{\|\boldsymbol{X}\|_* \le \mathcal{X} \\ \|\mathrm{P}_{\boldsymbol{A}^\perp \boldsymbol{B}^\perp}(\bar{\boldsymbol{X}})\|_* \le \mathcal{P}}} \frac{\mathcal{B}}{|\Omega|} \sum_{ij} E_{ij} \right]$$

$$\le \mathbb{E}_\sigma \left[ \sup_{\substack{\|\boldsymbol{X}\|_* \le \mathcal{X} \\ \|p_{AB}(\boldsymbol{X})\|_* \le \mathcal{P}}} \frac{\mathcal{B}}{|\Omega|} \|\boldsymbol{E}\|_1 \right]. \tag{28}$$

For the second term in (27), using the Rademacher contraction principle, we can bound it by:

$$\mathbb{E}_{\sigma}\left[\sup_{\substack{\|\boldsymbol{X}\|_* \leq \mathcal{X} \\ \|\mathrm{P}_{\boldsymbol{A}^{\perp}\boldsymbol{B}^{\perp}}(\boldsymbol{X})\|_* \leq \mathcal{P}}} \frac{1}{|\Omega|}\sum_{i,j}\Gamma_{ij}F_{ij}\right] = \mathbb{E}_{\sigma}\left[\sup_{\substack{\|\boldsymbol{X}\|_* \leq \mathcal{X} \\ \|\mathrm{P}_{\boldsymbol{A}^{\perp}\boldsymbol{B}^{\perp}}(\boldsymbol{X})\|_* \leq \mathcal{P}}} \frac{L_\ell}{|\Omega|}\langle\boldsymbol{\Gamma}, \boldsymbol{F}\rangle\right]. \tag{29}$$

Using Hölder's inequality, the right-hand side (R.H.S.) of (29) can be bounded by

$$\mathbb{E}_{\sigma}\left[\sup_{\substack{|\boldsymbol{X}\|_* \leq \mathcal{X} \\ \|\mathrm{P}_{\boldsymbol{A}^{\perp}\boldsymbol{B}^{\perp}}(\boldsymbol{X})\|_* \leq \mathcal{P}}} \frac{L_\ell}{|\Omega|}\langle\boldsymbol{\Gamma}, \boldsymbol{F}\rangle\right] \leq \mathbb{E}_{\sigma}\left[\sup_{\substack{\|\boldsymbol{X}\|_* \leq \mathcal{X} \\ \|\mathrm{P}_{\boldsymbol{A}^{\perp}\boldsymbol{B}^{\perp}}(\boldsymbol{X})\|_* \leq \mathcal{P}}} \frac{L_\ell}{|\Omega|}\|\boldsymbol{\Gamma}\|_2\|\boldsymbol{F}\|_*\right]. \tag{30}$$

According to (28) and (30), the Rademacher complexity can be bounded as follows:

$$\mathfrak{R}_\ell(\mathcal{F}) \leq \mathbb{E}_{\sigma}\left[\sup_{\substack{\|\boldsymbol{X}\|_* \leq \mathcal{X} \\ \|\mathrm{P}_{\boldsymbol{A}^{\perp}\boldsymbol{B}^{\perp}}(\boldsymbol{X})\|_* \leq \mathcal{P}}} \frac{\mathcal{B}}{|\Omega|}\|\boldsymbol{E}\|_1\right] + \mathbb{E}_{\sigma}\left[\sup_{\substack{\|\boldsymbol{X}\|_* \leq \mathcal{X} \\ \|\mathrm{P}_{\boldsymbol{A}^{\perp}\boldsymbol{B}^{\perp}}(\boldsymbol{X})\|_* \leq \mathcal{P}}} \frac{L_\ell}{|\Omega|}\|\boldsymbol{\Gamma}\|_2\|\boldsymbol{F}\|_*\right]. \tag{31}$$

Based on Lemma 10 and Lemma 11 in (Shamir & Shalev-Shwartz, 2014) and Definition B.2, the expression in (31) can be further bounded by:

$$\mathfrak{R}_\ell(\mathcal{F}) \leq \frac{\mathcal{B}}{\sqrt{t}} + \frac{2CL_\ell\sqrt{t}(\sqrt{m} + \sqrt{n})}{|\Omega|}. \tag{32}$$

Then, we set $t$ as the optimal minimizer for the R.H.S. of (32), i.e., $t = \frac{|\Omega|\mathcal{B}}{2CL_\ell\sqrt{t}(\sqrt{m}+\sqrt{n})}$, we get

$$\mathbb{E}_{\Omega}[\mathfrak{R}_\ell(\mathcal{F})] \leq \sqrt{\frac{9Cl_\ell\mathcal{X}\mathcal{B}(\sqrt{m} + \sqrt{n})}{|\Omega|}}. \tag{33}$$

In summary, after combining (26) and (33), we have

$$\mathbb{E}_{\Omega}[\mathfrak{R}_\ell(\mathcal{F})] \leq \min\left\{2L_\ell\mathcal{Q}\mathcal{S}\mathcal{P}\sqrt{\frac{\log 2N}{|\Omega|}}, \sqrt{\frac{9Cl_\ell\mathcal{X}\mathcal{B}(\sqrt{m} + \sqrt{n})}{|\Omega|}}\right\},$$

which concludes the proof of Lemma 3.2. □

## C. Extending the Generalization Bound to MSE

The generalization error bound established in Theorem 3.3 is independent of the sampling distribution. In this part, we show how this distribution-free bound can be extended to the mean squared error (MSE), which is widely used in matrix completion.

In Theorem 3.3, we provide a generalization error bound that holds under any sampling distributions. It is important to note that the definition of the generalization error $\mathcal{L}(\boldsymbol{X})$ depends on the given sampling distribution, i.e.,

$$\mathcal{L}(\boldsymbol{X}) = \mathbb{E}_{(i,j)\sim p}\left[\ell(X_{ij}, R_{ij})\right].$$

In particular, for squared loss under uniform sampling, $\mathcal{L}(\boldsymbol{X})$ corresponds to the MSE:

$$\mathrm{MSE}(\boldsymbol{X}) = \mathbb{E}_{(i,j)\sim U}\left[(X_{ij} - R_{ij})^2\right].$$

To clarify the connection between the bound of $\mathcal{L}(\boldsymbol{X})$ in Theorem 3.3 and MSE—a key evaluation metric in matrix completion—we have the following analysis. Under the squared loss, one can find that the difference between $\mathcal{L}(\boldsymbol{X})$ and MSE is that $\mathcal{L}(\boldsymbol{X})$ is expectation under an arbitrary sampling distribution $p$, while MSE assumes the uniform distribution $U$. From Theorem 3.3, we denote the bound of $\mathcal{L}(\boldsymbol{X})$ as $W$, then

$$\mathcal{L}(\boldsymbol{X}) = \mathbb{E}_{(i,j)\sim p}\left[(X_{ij} - R_{ij})^2\right] \leq W.$$

Applying the total variation distance bound on expectation for discrete distributions, we obtain:

$$\mathbb{E}_{(i,j)\sim U}\left[(X_{ij} - R_{ij})^2\right] \leq \mathbb{E}_{(i,j)\sim p}\left[(X_{ij} - R_{ij})^2\right] + 2M \cdot TV(U,p) \leq W + 2M \cdot TV(U,p),$$

where $M$ is the upper bound of $(X_{ij} - R_{ij})^2$, and $TV(U,p)$ represents the total variation distance between the uniform and arbitrary sampling distributions. This result extends our bound to MSE under arbitrary sampling distributions.

## D. Discussion about the Number of Observations

We first provide a detailed derivation of Corollary 3.4. Recalling the Theorem 3.3, we have

$$\mathcal{L}(\boldsymbol{X}) \leq \hat{\mathcal{L}}(\boldsymbol{X}) + \mathcal{B}\sqrt{\frac{\log(1/\delta)}{2|\Omega|}} + 2\min\left\{2L_\ell \mathcal{QSP}\sqrt{\frac{\log 2N}{|\Omega|}}, \sqrt{\frac{9CL_\ell \mathcal{XB}(\sqrt{m}+\sqrt{n})}{|\Omega|}}\right\}.$$

Considering the $\epsilon$-error recovery, i.e.

$$\mathcal{L}(\boldsymbol{X}) = \mathbb{E}_{(i,j)}\left[\ell(f(i,j), R_{ij})\right] \leq \epsilon,$$

for any $\epsilon > 0$, then we let

$$\mathcal{L}(\boldsymbol{X}) \leq \hat{\mathcal{L}}(\boldsymbol{X}) + \mathcal{B}\sqrt{\frac{\log(1/\delta)}{2|\Omega|}} + 2\min\left\{2L_\ell \mathcal{QSP}\sqrt{\frac{\log 2N}{|\Omega|}}, \sqrt{\frac{9CL_\ell \mathcal{XB}(\sqrt{m}+\sqrt{n})}{|\Omega|}}\right\} \leq \epsilon. \tag{34}$$

As discussed in Section 3, the empirical error is a relatively well-controlled quantity. Therefore, we do not consider the impact of this term here. Then (34) can be expressed as follows:

$$\mathcal{B}\sqrt{\frac{\log(1/\delta)}{2|\Omega|}} + 2\min\left\{2L_\ell \mathcal{QSP}\sqrt{\frac{\log 2N}{|\Omega|}}, \sqrt{\frac{9CL_\ell \mathcal{XB}(\sqrt{m}+\sqrt{n})}{|\Omega|}}\right\} \leq \epsilon. \tag{35}$$

After separating the term $\Omega$ in (35), we have

$$\mathcal{B}\sqrt{\frac{\log(1/\delta)}{2\epsilon^2}} + 2\min\left\{2L_\ell \mathcal{QSP}\sqrt{\frac{\log 2N}{\epsilon^2}}, \sqrt{\frac{9CL_\ell \mathcal{XB}(\sqrt{m}+\sqrt{n})}{\epsilon^2}}\right\} \leq \sqrt{|\Omega|}. \tag{36}$$

We handle the two terms in the minimum of (36) separately. As an example, we only consider the first term. For the second term, the approach is the same as the first term. Specifically, we have the following

$$\mathcal{B}\sqrt{\frac{\log(1/\delta)}{2\epsilon^2}} + 4L_\ell \mathcal{QSP}\sqrt{\frac{\log 2N}{\epsilon^2}} \leq \sqrt{|\Omega|}.$$

Taking the square of both sides of the above expressing gives

$$\left(\mathcal{B}\sqrt{\frac{\log(1/\delta)}{2\epsilon^2}} + 4L_\ell \mathcal{QSP}\sqrt{\frac{\log 2N}{\epsilon^2}}\right)^2 \leq |\Omega|. \tag{37}$$

Then, expanding the square operation in (37) yields

$$\mathcal{B}^2 \frac{\log(1/\delta)}{2\epsilon^2} + 2\mathcal{B} \cdot 4L_\ell \mathcal{QSP}\sqrt{\frac{\log(1/\delta)}{2\epsilon^2}}\sqrt{\frac{\log 2N}{\epsilon^2}} + 16L_\ell^2 \mathcal{Q}^2 \mathcal{S}^2 \mathcal{P}^2 \frac{\log 2N}{\epsilon^2} \leq |\Omega|. \tag{38}$$

By ignoring the constant terms in (38) that are independent of the matrix size $N$ and lower-order terms, we can derive the following conclusion for the number of observations:

$$|\Omega| = O\left(\mathcal{P}^2 log N\right). \tag{39}$$

Similarly, by handling the second term in (36), we obtain the following:

$$|\Omega| = O\left(\frac{\mathcal{X}^2 \sqrt{N}}{\epsilon^2}\right). \tag{40}$$

Combining (39) and (40), we draw the conclusion in Corollary 3.4:

$$|\Omega| = \min\left\{O\left(\frac{\mathcal{P}^2 \log N}{\epsilon^2}\right), O\left(\frac{\mathcal{X}^2 \sqrt{N}}{\epsilon^2}\right)\right\}.$$

We provide a detailed discussion of Corollary 3.4 under different side-information scenarios. For simplicity in our analysis, we focus on one of the two lower bounds, as this is sufficient to demonstrate the effectiveness of our model.

- When the side information is perfect, we focus on the first lower bound. $\mathcal{P}$ remains invariant with matrix dimension $N$, then the sample complexity required is $O(\log N)$. This result is consistent with the result of (Xu et al., 2013), where it is proved that $O(\log N)$ observations are required for the matrix completion task by solving IMC. However, in (Xu et al., 2013), it is assumed that the observed positions are uniformly distributed, which facilitates the analysis using convex optimization techniques. In our proof, we leverage theories from PAC learning, thereby avoiding any assumptions regarding the distribution of the observed positions.

- When we have incomplete side-information, we focus on the first lower bound. The sample complexity required is $O(\mathcal{P}^2 \log N)$, where $\mathcal{P}$ decreases as the completeness increases. Moreover the sample complexity decreases quadratically with the value of $\mathcal{P}$.

- When we have no side information, we focus on the second lower bound. We get $\mathcal{X} = O(\sqrt{N})$ as a reasonable assumption (Shamir & Shalev-Shwartz, 2014), then the entire sample complexity is $O(N^{2/3})$, which is consistent with the conclusion of standard matrix completion without any assumption of sampling distribution (Shamir & Shalev-Shwartz, 2014).

It is evident that our conclusions apply to the more general matrix completion problems existing in the literature, for the reason that our model differs from traditional decomposition-based models (Yang et al., 2020; Chiang et al., 2015). We focus not only on the components after decomposition but also on the overall low-rank property of the matrix.

## E. The details of Algorithm 1

**The Detailed Derivation of Closed-Form Solutions:** In this section, we provide a detailed derivation of the closed-form solutions for Algorithm 1. Recalling (14) in Section 4

$$\mathfrak{L}(\boldsymbol{X}, \boldsymbol{Y}, \boldsymbol{M}_1, \boldsymbol{M}_2, \beta) = \|\boldsymbol{X}\|_* + \lambda\|\boldsymbol{Y}\|_* + \frac{\beta}{2}\|\mathrm{P}_\Omega(\boldsymbol{X}) - \mathrm{P}_\Omega(\boldsymbol{R})\|_F^2 + \frac{\beta}{2}\|\boldsymbol{Y} - \mathrm{P}_{\boldsymbol{A}^\perp \boldsymbol{B}^\perp}(\boldsymbol{X})\|_F^2$$
$$+ \langle \boldsymbol{M}_1, \mathrm{P}_\Omega(\boldsymbol{X}) - \mathrm{P}_\Omega(\boldsymbol{R}) \rangle + \langle \boldsymbol{M}_2, \boldsymbol{Y} - \mathrm{P}_{\boldsymbol{A}^\perp \boldsymbol{B}^\perp}(\boldsymbol{X}) \rangle,$$

then we solve $\boldsymbol{X}$ in (15) and $\boldsymbol{Y}$ in (16) separately.

Update $\boldsymbol{X}$:

$$\boldsymbol{X}^{k+1} = \arg\min_{\boldsymbol{X}} \mathfrak{L}(\boldsymbol{X}, \boldsymbol{Y}^k, \boldsymbol{M}_1^k, \boldsymbol{M}_2^k, \beta)$$

$$= \arg\min_{\boldsymbol{X}} \|\boldsymbol{X}\|_* + \frac{\beta}{2}\|\mathrm{P}_\Omega(\boldsymbol{X}) - \mathrm{P}_\Omega(\boldsymbol{R})\|_F^2 + \frac{\beta}{2}\|\boldsymbol{Y}^k - \mathrm{P}_{\boldsymbol{A}^\perp \boldsymbol{B}^\perp}(\boldsymbol{X})\|_F^2 + \langle \boldsymbol{M}_1^k,$$

$$\mathrm{P}_\Omega(\boldsymbol{X}) - \mathrm{P}_\Omega(\boldsymbol{R}) \rangle + \langle \boldsymbol{M}_2^k, \boldsymbol{Y}^k - \mathrm{P}_{\boldsymbol{A}^\perp \boldsymbol{B}^\perp}(\boldsymbol{X}) \rangle.$$

Adding constant terms $\frac{1}{2\beta}\|\boldsymbol{M}_1^k\|_F^2$ and $\frac{1}{2\beta}\|\boldsymbol{M}_2^k\|_F^2$ and completing the square, we have:

$$\boldsymbol{X}^{k+1} = \arg\min_{\boldsymbol{X}} \|\boldsymbol{X}\|_* + \frac{\beta}{2}\|\mathrm{P}_\Omega(\boldsymbol{X}) - \mathrm{P}_\Omega(\boldsymbol{R})\|_F^2 + \langle \boldsymbol{M}_1^k, \mathrm{P}_\Omega(\boldsymbol{X}) - \mathrm{P}_\Omega(\boldsymbol{R}) \rangle + \frac{1}{2\beta}\|\boldsymbol{M}_1^k\|_F^2$$

$$+ \frac{\beta}{2}\|\boldsymbol{Y}^k - \mathrm{P}_{\boldsymbol{A}^\perp \boldsymbol{B}^\perp}(\boldsymbol{X})\|_F^2 + \langle \boldsymbol{M}_2^k, \boldsymbol{Y}^k - \mathrm{P}_{\boldsymbol{A}^\perp \boldsymbol{B}^\perp}(\boldsymbol{X}) \rangle + \frac{1}{2\beta}\|\boldsymbol{M}_2^k\|_F^2$$

$$= \arg\min_{\boldsymbol{X}} \|\boldsymbol{X}\|_* + \frac{\beta}{2}\|\mathrm{P}_\Omega(\boldsymbol{X} - \boldsymbol{R}) + \frac{\boldsymbol{M}_1^k}{\beta}\|_F^2 + \frac{\beta}{2}\|\mathrm{P}_{\boldsymbol{A}^\perp \boldsymbol{B}^\perp}(\boldsymbol{X}) - \boldsymbol{Y}^k + \frac{\boldsymbol{M}_2^k}{\beta}\|_F^2. \tag{41}$$

The presence of both nuclear norm terms and quadratic terms in the objective function complicates the overall problem, making it challenging to achieve convergence through iterative methods. However, in practical implementations, we often do not require each iterative step to be solved to high precision for the algorithm to remain viable. Building on this principle, we propose approximating the subproblems by linearizing the quadratic terms of the objective function.

Specifically, we use the following approximate strategy to handle $\|\mathrm{P}_\Omega(\boldsymbol{X} - \boldsymbol{R}) + \frac{\boldsymbol{M}_1^k}{\beta}\|_F^2$ and $\|\mathrm{P}_{\boldsymbol{A}^\perp \boldsymbol{B}^\perp}(\boldsymbol{X}) - \boldsymbol{Y}^k + \frac{\boldsymbol{M}_2^k}{\beta}\|_F^2$:

$$f(\boldsymbol{X}) \approx f(\boldsymbol{X}^k) + \langle g^k, \boldsymbol{X} - \boldsymbol{X}^k \rangle + \frac{1}{\tau}\|\boldsymbol{X} - \boldsymbol{X}^k\|_F^2.$$

where $f$ is differentiable at $\boldsymbol{X}^k$, $g^k$ is the gradient at $\boldsymbol{X}^k$, and $\tau > 0$ is the proximal parameter.

Applying the approximate strategy for the second and third parts of (41), we have:

$$\|\mathrm{P}_\Omega(\boldsymbol{X} - \boldsymbol{R}) + \frac{\boldsymbol{M}_1^k}{\beta}\|_F^2 \approx \|\mathrm{P}_\Omega(\boldsymbol{X}^k - \boldsymbol{R}) + \frac{\boldsymbol{M}_1^k}{\beta}\|_F^2 + 2 < \mathrm{P}_\Omega(\boldsymbol{X}^k - \boldsymbol{R} + \frac{\boldsymbol{M}_1^k}{\beta}), \boldsymbol{X} - \boldsymbol{X}^k > + \frac{1}{\tau}\|\boldsymbol{X} - \boldsymbol{X}^k\|_F^2,$$

$$\|\mathrm{P}_{\boldsymbol{A}^\perp \boldsymbol{B}^\perp}(\boldsymbol{X}) - \boldsymbol{Y}^k + \frac{\boldsymbol{M}_2^k}{\beta}\|_F^2 \approx \|\mathrm{P}_{\boldsymbol{A}^\perp \boldsymbol{B}^\perp}(\boldsymbol{X}^k) - \boldsymbol{Y}^k + \frac{\boldsymbol{M}_2^k}{\beta}\|_F^2 + 2 < \mathrm{P}_{\boldsymbol{A}^\perp \boldsymbol{B}^\perp}(\boldsymbol{X}^k - \boldsymbol{Y}^k + \frac{\boldsymbol{M}_2^k}{\beta}), \boldsymbol{X} - \boldsymbol{X}^k >$$

$$+ \frac{1}{\tau}\|\boldsymbol{X} - \boldsymbol{X}^k\|_F^2.$$

Substituting the results back to (41), we obtain:

$$\boldsymbol{X}^{k+1} \approx \arg\min_{\boldsymbol{X}} \|\boldsymbol{X}\|_* + \frac{\beta}{2}\|\mathrm{P}_\Omega(\boldsymbol{X}^k - \boldsymbol{R}) + \frac{\boldsymbol{M}_1^k}{\beta}\|_F^2 + \beta < \mathrm{P}_\Omega(\boldsymbol{X}^k - \boldsymbol{R} + \frac{\boldsymbol{M}_1^k}{\beta}), \boldsymbol{X} - \boldsymbol{X}^k > + \frac{\beta}{2\tau}\|\boldsymbol{X} - \boldsymbol{X}^k\|_F^2$$

$$+ \frac{\beta}{2}\|\mathrm{P}_{\boldsymbol{A}^\perp \boldsymbol{B}^\perp}(\boldsymbol{X}^k) - \boldsymbol{Y}^k + \frac{\boldsymbol{M}_2^k}{\beta}\|_F^2 + \beta < \mathrm{P}_{\boldsymbol{A}^\perp \boldsymbol{B}^\perp}(\boldsymbol{X}^k - \boldsymbol{Y}^k + \frac{\boldsymbol{M}_2^k}{\beta}), \boldsymbol{X} - \boldsymbol{X}^k > + \frac{\beta}{2\tau}\|\boldsymbol{X} - \boldsymbol{X}^k\|_F^2$$

$$= \arg\min_{\boldsymbol{X}} \|\boldsymbol{X}\|_* + \beta < \mathrm{P}_\Omega(\boldsymbol{X}^k - \boldsymbol{R} + \frac{\boldsymbol{M}_1^k}{\beta}) + \mathrm{P}_{\boldsymbol{A}^\perp \boldsymbol{B}^\perp}(\boldsymbol{X}^k - \boldsymbol{Y}^k + \frac{\boldsymbol{M}_2^k}{\beta}), \boldsymbol{X} - \boldsymbol{X}^k > + \frac{\beta}{\tau}\|\boldsymbol{X} - \boldsymbol{X}^k\|_F^2.$$

By adding constant and completing square, we have:

$$\boldsymbol{X}^{k+1} \approx \arg\min_{\boldsymbol{X}} \|\boldsymbol{X}\|_* + \frac{\beta}{\tau}\|\boldsymbol{X} - \boldsymbol{X}^k + \frac{\tau}{2}\mathrm{P}_\Omega(\boldsymbol{X}^k - \boldsymbol{R} + \frac{\boldsymbol{M}_1^k}{\beta}) + \frac{\tau}{2}\mathrm{P}_{\boldsymbol{A}^\perp \boldsymbol{B}^\perp}(\boldsymbol{X}^k - \boldsymbol{Y}^k + \frac{\boldsymbol{M}_2^k}{\beta})\|_F^2$$

$$= \arg\min_{\boldsymbol{X}} \frac{\tau}{2\beta}\|\boldsymbol{X}\|_* + \frac{1}{2}\|\boldsymbol{X} - \boldsymbol{X}^k + \frac{\tau}{2}\mathrm{P}_\Omega(\boldsymbol{X}^k - \boldsymbol{R} + \frac{\boldsymbol{M}_1^k}{\beta}) + \frac{\tau}{2}\mathrm{P}_{\boldsymbol{A}^\perp \boldsymbol{B}^\perp}(\boldsymbol{X}^k - \boldsymbol{Y}^k + \frac{\boldsymbol{M}_2^k}{\beta})\|_F^2. \quad (42)$$

To solve this problem in (42), we utilize the following lemma.

**Lemma E.1.** *(Cai et al., 2010; Ma et al., 2011b)* *Given $\boldsymbol{M} \in \mathbb{R}^{m \times n}$ and $\delta > 0$, let $\boldsymbol{M} = \boldsymbol{U}\boldsymbol{\Sigma}\boldsymbol{V}^T$ be the SVD of $\boldsymbol{M}$ and $\boldsymbol{I}$ be the identity matrix. We define the following "singular value shrinkage" operator:*

$$S_\delta(\boldsymbol{M}) := \boldsymbol{U}(\boldsymbol{\Sigma} - \delta\boldsymbol{I})^+ \boldsymbol{V}^T,$$

*where $(a)^+ := \max\{a, 0\}$. Then, it can be shown that*

$$S_\delta(\boldsymbol{M}) = \arg\min_{\boldsymbol{N}} \delta\|\boldsymbol{N}\|_* + \frac{1}{2}\|\boldsymbol{N} - \boldsymbol{M}\|_F^2.$$

Applying Lemma E.1 to (42), we have the closed form solution for updating $\boldsymbol{X}^{k+1}$:

$$\boldsymbol{X}^{k+1} = S_{\frac{\tau}{2\beta}}\left(\boldsymbol{X}^k - \frac{\tau}{2}\mathrm{P}_\Omega(\boldsymbol{X}^k - \boldsymbol{R} + \frac{\boldsymbol{M}_1^k}{\beta}) - \frac{\tau}{2}\mathrm{P}_{\boldsymbol{A}^\perp \boldsymbol{B}^\perp}(\boldsymbol{X}^k - \boldsymbol{Y}^k + \frac{\boldsymbol{M}_2^k}{\beta})\right).$$

For the update of $\boldsymbol{Y}$,

$$\boldsymbol{Y}^{k+1} = \arg\min_{\boldsymbol{Y}}\lambda\|\boldsymbol{Y}\|_* + \langle \boldsymbol{M}_2^k, \mathrm{P}_{\boldsymbol{A}^\perp \boldsymbol{B}^\perp}(\boldsymbol{X}^{k+1}) - \boldsymbol{Y}\rangle + \frac{\beta}{2}\|\mathrm{P}_{\boldsymbol{A}^\perp \boldsymbol{B}^\perp}(\boldsymbol{X}^{k+1}) - \boldsymbol{Y}\|_F^2.$$

Adding constant terms and completing square, we have

$$\boldsymbol{Y}^{k+1} = \arg\min_{\boldsymbol{Y}} \frac{\lambda}{\beta}\|\boldsymbol{Y}\|_* + \frac{1}{2}\|\boldsymbol{Y} - \mathrm{P}_{\boldsymbol{A}^\perp \boldsymbol{B}^\perp}(\boldsymbol{X}^{k+1}) - \frac{\boldsymbol{M}_2^k}{\beta}\|_F^2.$$

The update for $\boldsymbol{Y}^{k+1}$ is considerably simpler than that for $\boldsymbol{X}^{k+1}$, and thus we do not require linearization. By applying Lemma E.1, we have

$$\boldsymbol{Y}^{k+1} = S_{\frac{\lambda}{\beta}}\left(\mathrm{P}_{\boldsymbol{A}^\perp \boldsymbol{B}^\perp}(\boldsymbol{X}^{k+1}) + \frac{\boldsymbol{M}_2^k}{\beta}\right).$$

For the regularization parameter $\beta$, we adopt an incremental strategy to accelerate algorithm convergence. Specifically, we define a growth factor $\rho$ and an upper limit $\beta_{\max}$. During each iteration, $\beta$ is updated as $\beta = \rho \cdot \beta$ until $\beta$ reaches $\beta_{\max}$, at which point the growth stops. The updates of $M_1$ and $M_2$ follow the standard update rules in the ADMM algorithm.

In summary, we have:

$$\boldsymbol{X}^{k+1} = S_{\frac{\tau}{2\beta}}\left(\boldsymbol{X}^k - \frac{\tau}{2}\mathrm{P}_\Omega(\boldsymbol{X}^k - \boldsymbol{R} + \frac{\boldsymbol{M}_1^k}{\beta}) - \frac{\tau}{2}\mathrm{P}_{\boldsymbol{A}^\perp \boldsymbol{B}^\perp}(\boldsymbol{X}^k - \boldsymbol{Y}^k + \frac{\boldsymbol{M}_2^k}{\beta})\right),$$

$$\boldsymbol{Y}^{k+1} = S_{\frac{\lambda}{\beta}}\left(\mathrm{P}_{\boldsymbol{A}^\perp \boldsymbol{B}^\perp}(\boldsymbol{X}^{k+1}) + \frac{\boldsymbol{M}_2^k}{\beta}\right),$$

$$\boldsymbol{M}_1^{k+1} = \boldsymbol{M}_1^k + \beta^k(\mathrm{P}_\Omega(\boldsymbol{X}^{k+1}) - \mathrm{P}_\Omega(\boldsymbol{R})),$$

$$\boldsymbol{M}_2^{k+1} = \boldsymbol{M}_2^k + \beta^k(\boldsymbol{Y}^{k+1} - \mathrm{P}_{\boldsymbol{A}^\perp \boldsymbol{B}^\perp}(\boldsymbol{X}^{k+1})),$$

$$\beta^{k+1} = \min\{\beta_{\max}, \rho\beta^k\}.$$

Therefore, we have completed the detailed form of Algorithm 1.

**Computational Complexity of Algorithm 1:** For a target matrix $\boldsymbol{R} \in \mathbb{R}^{m \times n}$ and side information $\boldsymbol{A} \in \mathbb{R}^{m \times d}$, $\boldsymbol{B} \in \mathbb{R}^{n \times d}$

(for convenience, here we assume $r_A = r_B = d$), the per-iteration complexity of the linear-ADMM algorithm for OCMC consists of three parts:

- Calculating $P_{A^\perp B^\perp}(X) : O(\min(dmn + dm^2, dmn + dn^2))$.

- Singular value shrinkage: $O(\min(m^2 n, n^2 m))$.

- Matrix inner product: $O(mn)$.

The total complexity is dominated by the full SVD in singular value shrinkage step, i.e., $O(\min(m^2 n, n^2 m))$. For large-scale problems, we can replace the full SVD with more efficient methods, such as randomized SVD and Lanczos method, whose complexities are $O(mn + r^2 m + r^3)$ and $O(rmn)$, respectively, where $r$ is rank of $\boldsymbol{R}$.

## F. More Experiments

To further validate the effectiveness, robustness, and generalizability of the proposed OCMC model, we present several additional experiments in this appendix. These include analyses of the relationship between completion accuracy and the completeness of side information, the sampling rate required to achieve high-accuracy completion, the impact of noisy side information on performance, the comparison between OCMC and DirtyIMC under fully complete side information, and a comprehensive evaluation on multi-label learning datasets. All synthetic data are generated following the setup described in Section 5 of the main text, and all reported results are averaged over 10 independent runs to ensure statistical reliability.

**Additional Results for Figure 2:** To further support the observations in Figure 2, we conduct additional experiments using matrices with different ranks. The nuclear norms of $P_{\boldsymbol{A}^\perp \boldsymbol{B}^\perp}(\boldsymbol{R})$ and $P_{\boldsymbol{A}^\perp \boldsymbol{B}^\perp}(\boldsymbol{X})$ are summarized in Table 5.

*Table 5.* Nuclear norm of orthogonal complement under different completeness levels of side information.

| Completeness level | 0 | 0.2 | 0.4 | 0.6 | 0.8 | 1 |
|---|---|---|---|---|---|---|
| $r = 5$ | | | | | | |
| Target matrix | $1 \pm 0$ | $0.66 \pm 0.05$ | $0.40 \pm 0.03$ | $0.22 \pm 0.03$ | $0.08 \pm 0.01$ | $0 \pm 0$ |
| Random matrix | $1 \pm 0$ | $0.99 \pm 0.004$ | $0.98 \pm 0.007$ | $0.97 \pm 0.005$ | $0.96 \pm 0.013$ | $0.94 \pm 0.01$ |
| $r = 15$ | | | | | | |
| Target matrix | $1 \pm 0$ | $0.67 \pm 0.02$ | $0.48 \pm 0.02$ | $0.24 \pm 0.01$ | $0.07 \pm 0.01$ | $0 \pm 0$ |
| Random matrix | $1 \pm 0$ | $0.96 \pm 0.005$ | $0.94 \pm 0.004$ | $0.90 \pm 0.005$ | $0.87 \pm 0.007$ | $0.84 \pm 0.01$ |

As observed from Table 5, with the increase of completeness level, the nuclear norm of the target matrix shows a sharper decrease compared to the random matrix, which is consistent with the results in Figure 2.

**Accuracy VS. Completeness:** In Figure 6, we evaluate the completion accuracy of the proposed OCMC method across different completeness levels under a fixed sampling rate. In this experiment, the target matrix is a $100 \times 100$ rank-10 matrix, and the sampling rate is fixed at 25%. Here, the DirtyIMC method is selected as the baseline for comparison.

From the results in Figure 6, we can observe that the completion error for both models decreases as the completeness of the side information increases. The error obtained by OCMC is consistently lower than that of DirtyIMC, indicating that the proposed OCMC can utilize incomplete side information more effectively.

**Sampling Rate for** 10%**-Error Completion:** In Table 6, we investigate the sampling rates required by different models to achieve a specific completion accuracy. It is worth noting that high accuracy is often crucial in applications such as high-precision positioning (Xiong et al., 2023) and some signal processing tasks (Zhou et al., 2023). In this experiment, we set the target completion accuracy to a relative error of 10% and compare the sampling rate needed by OCMC, DirtyIMC, and SVT under three completeness levels of side information: low (20%), medium (50%), and high (80%).

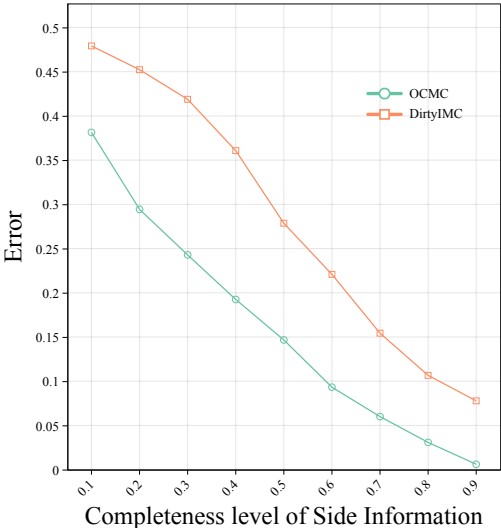

*Figure 6. Error VS. Completeness level of side information with 25% sampling rate.*

*Table 6.* Sampling rates required by different models to achieve 10%-error under varying completeness levels of side information.

| ACCURACY | COMPLETENESS | SVT | MAXIDE | DIRTYIMC | OCMC |
|---|---|---|---|---|---|
| 10% | LOW(20%) | 0.3903 | NONE | 0.3402 | 0.3168 |
| | MEDIUM(50%) | 0.3912 | NONE | 0.3135 | 0.2692 |
| | HIGH(80%) | 0.3980 | NONE | 0.2124 | 0.1853 |

As shown in Table 6, for any given completeness level, OCMC consistently requires the fewest samples to achieve the target 10% completion error, outperforming the benchmark methods. Specifically, OCMC uses approximately 4% fewer samples on average than DirtyIMC. In contrast, the SVT algorithm, which does not utilize side information, consistently demands more samples than both OCMC and DirtyIMC. Furthermore, the Maxide algorithm fails to reach the target accuracy across all settings, which is consistent with the findings in the synthetic experiments discussed in Section 5.

**Effect of Noisy Side Information:** To explore the robustness of the proposed OCMC under noisy side information cases, we conduct a experiment by adding different level of Gaussian noise into the incomplete side information, and compare the completion error. In our experiment, the target matrix is a $100 \times 100$ rank-10 matrix, and the completeness level is 50%. The side information matrices $A$ and $B$ are corrupted by additive noise matrices $E_A$ and $E_B$, whose entries are i.i.d. $\mathcal{N}(0, \alpha^2/m)$ and $\mathcal{N}(0, \alpha^2/n)$, respectively. The results are summarized in Table 7.

*Table 7.* Completion error with varying noise level under different sampling rates.

| NOISE LEVEL | $\alpha = 0$ | $\alpha = 0.1$ | $\alpha = 0.3$ | $\alpha = 0.5$ |
|---|---|---|---|---|
| SAMPLING RATE=0.1 | 0.689 | 0.697 | 0.753 | 0.842 |
| SAMPLING RATE=0.15 | 0.515 | 0.523 | 0.601 | 0.751 |
| SAMPLING RATE=0.2 | 0.319 | 0.324 | 0.422 | 0.551 |

As shown in Table 7, OCMC can still be adapted to such scenarios and maintain reasonable performance by adjusting the parameter $\lambda$ in (7) appropriately. In such cases, as the noise level increases, the side information becomes less helpful for matrix completion. For the extreme case when the side information is severely corrupted, we can set $\lambda = 0$, and the OCMC model will reduce to the standard matrix completion.

**Performance under Complete Side Information:** In Table 8, we compare the completion errors of OCMC and DirtyIMC in recovering a $100 \times 100$ rank-10 matrix with fully complete side information. Since DirtyIMC is designed for scenarios with noisy side information, we assume that the side information matrices are corrupted by Gaussian noise. Specifically, the

noise matrices $E_A$ and $E_B$ have i.i.d. entries drawn from $\mathcal{N}(0, 0.1^2/m)$ and $\mathcal{N}(0, 0.1^2/n)$, respectively.

*Table 8.* Completion errors of OCMC and DirtyIMC with 100% complete side information.

| SAMPLING RATE | 0.10 | 0.15 | 0.20 | 0.25 | 0.30 | 0.35 | 0.40 |
|---|---|---|---|---|---|---|---|
| OCMC | 0.1359 | 0.0781 | 0.0310 | 0.0128 | 0.0083 | 0.0062 | 0.0010 |
| DIRTYIMC | 0.1384 | 0.0963 | 0.0329 | 0.0156 | 0.0103 | 0.0075 | 0.0032 |

As shown in Table 8, when the side information is complete, the proposed OCMC still outperforms DirtyIMC. However, the performance gap between DirtyIMC and OCMC is smaller compared to the case with incomplete side information (as illustrated in Figure 4). This is because, for the OCMC model with complete side information, the dominance of $\mathrm{P}_{\boldsymbol{A}^\perp \boldsymbol{B}^\perp}(\boldsymbol{X})$ among the last three components in Table 1 is reduced. On the other hand, the problem formulation of DirtyIMC is

$$\min_{\boldsymbol{M},\boldsymbol{N}} \|\boldsymbol{M}\|_* + \lambda\|\boldsymbol{N}\|_* \quad \text{s.t.} \quad \mathrm{P}_\Omega(\boldsymbol{A}\boldsymbol{M}\boldsymbol{B}^\top + \boldsymbol{N}) = \mathrm{P}_\Omega(\boldsymbol{R}),$$

where the matrix $\boldsymbol{N}$ represents the sum of the last three components in Table 1. This formulation treats all three components equally. Although the contribution of $\mathrm{P}_{\boldsymbol{A}^\perp \boldsymbol{B}^\perp}(\boldsymbol{X})$ in the OCMC formulation becomes less dominant with complete side information, OCMC still benefits from focusing on it, rather than treating all three components equally.

Notably, the OCMC model is primarily designed for scenarios with incomplete side information, which reflects most real-world situations as shown in Section 5.4. While OCMC shows reduced advantage over DirtyIMC under complete side information, it significantly improves completion accuracy and robustness when the side information is incomplete.

**Complete Table for Multi-Label Learning:** From the complete results in Table 9, it can be observed that, among a total of 55 experimental categories, the OCMC model achieved 43 first-place rankings and 11 second-place rankings. Moreover, 43 first-place rankings and 49 second-place rankings were obtained by the OCMC, FNNC and DirtyIMC models. This further highlights the significant potential of models that account for imperfect side information in multi-label learning scenarios.

*Table 9.* Comparison of algorithms on all dataset with varying sampling rates $\omega$.

| DATASET | $\omega\%$ | SVT | MAXIDE | DIRTYIMC | FPC | OCMC | FNNM |
|---|---|---|---|---|---|---|---|
| ARTS | 10% | 0.3500 | **0.5746** | 0.4591 | 0.3402 | 0.5249 | 0.5126 |
| | 30% | 0.4639 | 0.6435 | 0.6416 | 0.5012 | **0.7206** | 0.6825 |
| | 50% | 0.6053 | 0.6784 | 0.7648 | 0.6103 | **0.8358** | 0.7625 |
| | 70% | 0.7271 | 0.6991 | 0.8584 | 0.7592 | **0.9029** | 0.8326 |
| | 90% | 0.8695 | 0.7135 | 0.9266 | 0.8763 | **0.9596** | 0.9115 |
| BUSINESS | 10% | 0.4963 | **0.8614** | 0.6245 | 0.8423 | 0.7629 | 0.8425 |
| | 30% | 0.5806 | 0.8817 | 0.7622 | 0.8901 | **0.9159** | 0.8952 |
| | 50% | 0.6932 | 0.8953 | 0.8400 | 0.9335 | **0.9531** | 0.9312 |
| | 70% | 0.7840 | 0.9038 | 0.8942 | 0.9612 | **0.9751** | 0.9580 |
| | 90% | 0.9190 | 0.9092 | 0.9345 | 0.9724 | **0.9866** | 0.9769 |
| COMPUTERS | 10% | 0.3594 | **0.6600** | 0.4165 | 0.5721 | 0.5796 | 0.5782 |
| | 30% | 0.4602 | 0.7254 | 0.5642 | 0.6610 | **0.7626** | 0.7236 |
| | 50% | 0.5868 | 0.7586 | 0.6900 | 0.7381 | **0.8594** | 0.8023 |
| | 70% | 0.7271 | 0.7796 | 0.7973 | 0.8131 | **0.9229** | 0.9036 |
| | 90% | 0.8471 | 0.7941 | 0.8857 | 0.8863 | **0.9608** | 0.9236 |
| ENTERTAINMENT | 10% | 0.3623 | **0.6528** | 0.5274 | 0.3602 | 0.5867 | 0.5768 |
| | 30% | 0.5134 | 0.6758 | 0.6975 | 0.4822 | **0.7527** | 0.7126 |
| | 50% | 0.6158 | 0.7222 | 0.8058 | 0.6153 | **0.8634** | 0.8036 |
| | 70% | 0.7321 | 0.7566 | 0.8861 | 0.7352 | **0.9205** | 0.8623 |
| | 90% | 0.8392 | 0.7775 | 0.9433 | 0.8562 | **0.9598** | 0.9026 |
| EDUCATION | 10% | 0.3791 | **0.6001** | 0.4994 | 0.4152 | 0.5637 | 0.5523 |
| | 30% | 0.4919 | 0.6569 | 0.6767 | 0.5274 | **0.7170** | 0.6823 |
| | 50% | 0.5975 | 0.6870 | 0.7915 | 0.6322 | **0.8294** | 0.7625 |
| | 70% | 0.7429 | 0.7163 | 0.8787 | 0.7523 | **0.9056** | 0.5426 |
| | 90% | 0.8547 | 0.7280 | 0.9384 | 0.8463 | **0.9521** | 0.9036 |
| HEALTH | 10% | 0.4623 | **0.7472** | 0.5185 | 0.4891 | 0.6459 | 0.6437 |
| | 30% | 0.5409 | 0.7974 | 0.6800 | 0.6102 | **0.8225** | 0.7982 |
| | 50% | 0.6332 | 0.8207 | 0.7853 | 0.7143 | **0.9067** | 0.8523 |
| | 70% | 0.7521 | 0.8357 | 0.8663 | 0.8059 | **0.9483** | 0.8962 |
| | 90% | 0.8513 | 0.8469 | 0.9267 | 0.8853 | **0.9724** | 0.9470 |
| REFERENCE | 10% | 0.3152 | **0.6494** | 0.4899 | 0.3132 | 0.5510 | 0.5260 |
| | 30% | 0.4836 | 0.7285 | 0.6556 | 0.4317 | **0.7424** | 0.7132 |
| | 50% | 0.5985 | 0.7663 | 0.7678 | 0.5733 | **0.8436** | 0.8211 |
| | 70% | 0.7225 | 0.7907 | 0.8567 | 0.7082 | **0.9143** | 0.8652 |
| | 90% | 0.8385 | 0.8087 | 0.9280 | 0.8372 | **0.9501** | 0.9125 |
| RECREATION | 10% | 0.3574 | **0.5859** | 0.4847 | 0.3982 | 0.5301 | 0.5241 |
| | 30% | 0.4356 | 0.6694 | 0.6652 | 0.5152 | **0.7126** | 0.6782 |
| | 50% | 0.5726 | 0.7103 | 0.7852 | 0.6306 | **0.8271** | 0.7721 |
| | 70% | 0.7154 | 0.7364 | 0.8732 | 0.7532 | **0.9071** | 0.8623 |
| | 90% | 0.8352 | 0.7545 | 0.9360 | 0.8643 | **0.9520** | 0.9152 |
| SOCIAL | 10% | 0.3851 | **0.7472** | 0.4855 | 0.2365 | 0.6432 | 0.6105 |
| | 30% | 0.5149 | **0.7975** | 0.6508 | 0.3823 | 0.7744 | 0.7685 |
| | 50% | 0.6248 | 0.8207 | 0.7650 | 0.5309 | **0.8769** | 0.8362 |
| | 70% | 0.7261 | 0.8357 | 0.8552 | 0.6656 | **0.9356** | 0.8742 |
| | 90% | 0.8424 | 0.8469 | 0.9243 | 0.8212 | **0.9648** | 0.9322 |
| SCIENCE | 10% | 0.2736 | **0.5321** | 0.3970 | 0.2512 | 0.4625 | 0.4352 |
| | 30% | 0.4159 | 0.6338 | 0.5811 | 0.4283 | **0.6811** | 0.6251 |
| | 50% | 0.5624 | 0.6831 | 0.7173 | 0.5572 | **0.8019** | 0.7325 |
| | 70% | 0.7027 | 0.7153 | 0.8277 | 0.6971 | **0.8961** | 0.8251 |
| | 90% | 0.8374 | 0.7372 | 0.9124 | 0.8352 | **0.9443** | 0.8996 |
| SOCIETY | 10% | 0.3513 | **0.5989** | 0.4811 | 0.2752 | 0.5457 | 0.5412 |
| | 30% | 0.4523 | 0.6657 | 0.6474 | 0.4285 | **0.7152** | 0.6852 |
| | 50% | 0.5963 | 0.7036 | 0.7665 | 0.5606 | **0.8268** | 0.7652 |
| | 70% | 0.7024 | 0.7281 | 0.8572 | 0.6983 | **0.9010** | 0.8365 |
| | 90% | 0.8462 | 0.7459 | 0.9258 | 0.8256 | **0.9507** | 0.9021 |

