# OpenReview forum: "Matrix Completion with Incomplete Side Information via Orthogonal Complement Projection"
_ICML.cc/2025/Conference — ICML 2025 poster_

### Official Review · Reviewer_hL9t · 2025-03-08

**Overall Recommendation:** 3

**Summary:**

This paper studies the problem of matrix completion with side information. When the side information is not complete, the authors propose to use orthogonal complement projection to minimize the signals outside the side information, instead of constraining the recovered matrix in the space spanned by side information. The authors developed an ADMM algorithm to solve the proposed target with convergence guarantees. The theoretical investigation shows that the sample complexity decreases quadratically with the completeness level under the completeness level. The practical performance is examined on both synthetic and real data.

**Claims And Evidence:**

The claims are sound, and supported by theoretical investigations and experimental evidence.

**Essential References Not Discussed:**

No.

**Experimental Designs Or Analyses:**

In Figure 2, what's rationale of a decreasing trend in nuclear norm of a random matrix? It would be better to include an error bar, and vary ranks for a more complete message. It also helps motivate why minimizing part 4 (described in Figure 3) is effective.

How is matrix $B$ generated in Section 5.1? Does that follow a similar transformation like $A$?

**Methods And Evaluation Criteria:**

How are representative algorithms selected for comparison purpose? Is there a SOTA method for each dataset?

**Other Comments Or Suggestions:**

No

**Other Strengths And Weaknesses:**

No

**Questions For Authors:**

No

**Relation To Broader Scientific Literature:**

Matrix completion has a broad range of applications, such as collaborative filtering, computer vision, and recommendation systems.

**Theoretical Claims:**

I didn't go through the detailed proof, but the analysis is standard in PAC framework, and is sound to me.

The major question is about the interpretation of conditions. The assumptions on missing mechanism are unclear. The authors claim that the theory doesn't makes no assumptions regarding the distribution of observed entry positions. But it seems that a minimal observation probability among all entries is needed. It would be better to discuss how those conditions translate to requirements on missing mechanism and discuss applicability from there.

---

> ### Author Rebuttal · Authors · 2025-03-31
>
> Thank you for the constructive comments. We address all questions point by point below.
>
> **Q1: How are representative algorithms selected? Is there a SOTA method for each dataset?**
>
> A1: For each dataset, we emphasize the comparison within the scope of matrix completion methods. To ensure a comprehensive evaluation, we have carefully reviewed the existing literature and selected SOTA matrix completion algorithms as baselines, to the best of our knowledge.
> - Movielens-100k: FNNM [1] is recognized as a leading matrix completion method.
> - Multilabel learning: Both DirtyIMC [1,2] and FNNM demonstrate near-SOTA performance.
> - Effectiveness and Robustness validation: We compared against classical algorithms without side information (SVT, FPC) and Maxide from IMC framework.
>
> The selection of representative algorithms ensures that our comparisons are both meaningful and rigorous in addressing the core issue of matrix completion with incomplete side information.
>
> **Q2: Does generalization of $B$ in Section 5.1 follow a similar transformation like $A$?**
>
> A2: Yes, the generation of $B$ is similar to that of $A$. Specifically, $B=VQ$, following the similar transformation as $A=UT$ in Section 5.1, where $Q$ is a random matrix generated similarly to $T$.
>
> **Q3: In Fig 2, what's rationale of a decreasing trend in nuclear norm of a random matrix?**
>
> A3: The operator $P_{A^{\perp}B^{\perp}}$ satisfies the property  $||P_{A^{\perp}B^{\perp}}(X)||\_* \leq ||X||\_*$, as orthogonal projection typically reduces the nuclear norm.  Furthermore, as side information becomes more complete, the dimension of the projected subspace shrinks. Specifically, the subspace defined by $P\_{\check{A}^{\perp} \check{B}^{\perp}}(\cdot)$ is a subset of that by $P\_{{A}^{\perp} {B}^{\perp}}(\cdot)$,  leading to $||P_{\check{A}^{\perp} \check{B}^{\perp}}(X)||_\* \leq ||P\_{A^{\perp}B^{\perp}}(X)|| _\*  $, explaining the decrease trend.
>
> **Different Decreasing Rates of $X$ and $R$ in Fig 2**: For a random matrix, its row and column subspaces are uniformly distributed over the full space.  When a random matrix $X$ is projected by $P_{A^{\perp}B^{\perp}}$, where $A$ and $B$ are derived from the row and column subspaces of the target matrix $R$, the nuclear norm of projection decreases approximately at a rate of $1/n$.  In contrast, for $R$, since the projection is constructed based on its own subspaces, the decay is faster at approximately $1/r$.
>
> As suggested, we have added experiments with different ranks and the corresponding nuclear norm results are as follows.
> ||Completeness level|0|0.2|0.4|0.6|0.8|1|
> |-|-|-|-|-|-|-|-|
> |r=5|target matrix|1±0|0.66±0.05|0.40±0.03|0.22±0.03|0.08±0.01|0±0|
> |r=5|random matrix|1±0|0.99±0.004|0.98±0.007|0.97±0.005|0.96±0.013|0.94±0.01|
> |r=15|target matrix|1±0|0.67±0.02|0.48±0.02|0.24±0.01|0.07±0.01|0±0|
> |r=15|random matrix|1±0|0.96±0.005|0.94±0.004|0.90±0.005|0.87±0.007|0.84±0.01|
>
> As observed from the table above, with the increase of completeness level, the nuclear norm of the target matrix shows a sharper decrease compared to the random matrix.
> These results will be included in the final submission.
>
> **Q4: Assumption of Theorem 3.3 regarding the distribution of sampling distribution in page 6.**
>
> A4: Thanks for pointing out this confusion. The statement "our analysis makes no assumptions regarding the distribution of observed entry positions" means that Theorem 3.3 provides a generalization error bound that holds under any sampling distributions. It is important to note that the definition of the generalization error $L(X)$ depends on the given sampling distribution, i.e.,
> $$
> L(X)=E_{(i,j)\sim p}[l(X_{ij},R_{ij})].
> $$
> In particular, for squared loss under uniform sampling, $L(X)$ corresponds to the mean squared error (MSE):
> $$
> MSE(X)=E_{(i,j)\sim U}[(X_{ij}-R_{ij})^2].
> $$
> We believe the reviewer’s concern is how the distribution-free bound on $L(X)$ connects to MSE, given that MSE is a key metric in matrix completion. Under the squared loss, the difference between $L(X)$ and MSE is that $L(X)$ is expectation under an arbitrary sampling distribution $p$, while MSE assumes uniform. From Theorem 3.3, denote the bound of $L(X)$ as $W$, then
> $$
> L(X)=E_{(i,j)\sim p}[(X_{ij}-R_{ij})^2]\leq W.
> $$
> Applying the total variation distance bound on expectation for discrete distributions, we obtain:
> $$
> E_{(i,j)\sim U}[(X_{ij}-R_{ij})^2]\leq E_{(i,j)\sim p}[(X_{ij}-R_{ij})^2]+2M\cdot TV(U, p) \le W+2M\cdot TV(U, p)
> $$
> where $M$ is the upper bound of $(X_{ij}-R_{ij})^2$, and $TV(U, p)$ represents the total variation distance between the uniform and arbitrary sampling distributions. This result extends our bound to MSE under arbitrary sampling distributions.
>
> We hope that the above analysis and experimental results adequately address the reviewer’s comments.
>
> **References:**
> [1] Feature and nuclear norm minimization for matrix completion. IEEE TKDE, 2020.
> [2] Matrix completion with noisy side information. NIPS, 2015.

---

### Official Review · Reviewer_67wE · 2025-03-14

**Overall Recommendation:** 3

**Summary:**

In this work, the authors propose a new matrix completion method with incomplete side information. The incompleteness of the side information is defined and the solution called Orthogonal Complement Matrix Completion (OCMC) is developed. Theoretical analysis is given to show the upper bound of the errors. Experiments on various data verify the performance of the proposed OCMC method.

**Claims And Evidence:**

Yes.

**Essential References Not Discussed:**

No.

**Experimental Designs Or Analyses:**

Yes, I've checked the experimental results on synthetic data and real datasets (MLL and MovienLens). No big issues found.

**Methods And Evaluation Criteria:**

Yes.

**Other Comments Or Suggestions:**

N/A

**Other Strengths And Weaknesses:**

Strengths:

Proper problem definition and analysis on incomplete side information.

Weakness:

Lack of verification on the incompleteness of the side information in the real dataset.

**Questions For Authors:**

1. As Fig.2 shows, when the side information is complete. $\mathbb{P}_{A^\perp B^\perp}(\mathbf{R})$ will be zero. Thus, the second term in Eq.(7) seems to vanish, such that no side information is used. Could the author explain it?

2. In the synthetic data experiment (Fig.4 and 6), how about the performance when the completeness level of side Information is 100%? In this case, the side information is complete, so will the performance of OCMC be the same as dirtyIMC? If not, what causes the difference?

3. Two experiments on real dataset show the superior performance of the proposed method. However, there is a lack of analysis or evidence showing that the datasets match the hypotheses of incomplete side information. Therefore, it is unclear that the improvement on these datasets comes from properly dealing with the incomplete side information.

4. In the left column of line 426, page 8, 'Among them, dirtyIMC, FNNM, and OCMC, as models designed for incomplete side information....', it is confusing that dirtyIMC and FNNM only deal with imperfect(noisy) side information, not incomplete side information.

**Relation To Broader Scientific Literature:**

A new problem for matrix completion --- completion with 'incomplete side information' is defined and analyzed. It seems to be reasonable in real data, and the experiments also verify the performance (but lack of verifying whether the real data has incomplete side information)

**Theoretical Claims:**

I roughly check the correctness of the proof of Lemma 3.2 and related discussions about the number of observations. No big issues found.

---

> ### Author Rebuttal · Authors · 2025-03-31
>
> Thank you for the constructive questions. We address them point by point below.
>
> **Q1: The effect of $P_{A^\perp B^\perp}(X)$ in (7) when the side information is complete.**
>
>  A1: When the side information is complete, the column and row spaces of $R$ are fully contained in the column space of $A$ and row space of $B$, respectively, implying that $P_{A^\perp B^\perp}(R) = 0$. However, this does not mean that the second term in (7) vanishes. Instead, it will become more important in the optimization.
>
> In (7), $\lambda$ controls the regularization strength depending on the completeness of the side information. As discussed in Section 2.2, higher completeness → larger $\lambda$. When the side information is complete, $\lambda$ should be sufficiently large so that (7) becomes equivalent to a hard-constrained optimization:
> $$
> \min_{X}||X||\_*\quad s.t. \ P_{\Omega}(X) = P_{\Omega}(R), \ P_{A^\perp B^\perp}(X) = 0.
> $$
> In summary, when the side information is complete, the second term in (7) does not vanish but becomes stricter.
>
> **Q2: The performances of OCMC and DirtyIMC when the completeness level of side Information is 100\%.**
>
> A2:  We compare the completion errors of OCMC and DirtyIMC in recovering a $100 \times 100$ rank-$10$ matrix  with 100\% complete side Information. Since DirtyIMC is designed for scenarios with noisy side information,
> we assume that the side information matrices are corrupted by Gaussian noise.
>  Specifically, the noise matrices $E_A$ and $E_B$ have i.i.d. entries drawn from $\mathcal{N}(0, 0.1^2/m)$ and $\mathcal{N}(0, 0.1^2/n)$, respectively.
>
> |Observation rate|0.1|0.15|0.2|0.25|0.3|0.35|0.4|
> |-|-|-|-|-|-|-|-|
> |OCMC|0.1359|0.0781|0.0310|0.0128|0.0083|0.0062|0.0010|
> |DirtyIMC|0.1384|0.0963|0.0329|0.0156|0.0103|0.0075|0.0032
>
> As shown in the results, when the side information is complete, the proposed OCMC still outperforms DirtyIMC. However, the performance gap between DirtyIMC and OCMC is smaller compared to the case with incomplete side information (as illustrated in Figure 4). This is because, for the OCMC model with complete side information, the dominance of $P_{A^\perp B^\perp}(X)$ among the last three components in Table 1 of the manuscript is reduced.
> On the other hand, for dirtyIMC:
> $$
> \min\_{M} ||M||\_* + \lambda ||N||\_* \quad \text{s.t.} \quad P_{\Omega}(AMB^T + N) = P_{\Omega}(R),
> $$
> the matrix  $N$ represents the sum of the last three components in Table 1. This formulation treats all three components equally.  Although the contribution of  $P_{A^\perp B^\perp}(X)$ in OCMC formulation becomes less dominant with complete side information, OCMC still benefits from focusing on it, rather than treating all three components equally.
>
> It is worth noting that our OCMC model is primarily designed for scenarios with incomplete side information, which reflects most real-world scenarios—as demonstrated by our experiments. While OCMC shows reduced advantage over DirtyIMC under complete side information, it significantly improves recovery accuracy and robustness with incomplete side information is.
>
> **Q3: Analysis or evidence showing that the datasets match the hypotheses of incomplete side information.**
>
> A3: Thank you for raising this point. We justify the incomplete side information assumption from two perspectives:
>  -  Intuitive and practical perspective:  Taking the recommendation system as an example, side information typically consists of observable attributes of users (age/gender) and items (categories/genres). However, the complete information of user preferences or item characteristics is much richer.  It includes latent factors that are not captured but still contribute to the rating matrix. Intuitively, it is also unrealistic to assume that age, gender, or category alone fully describe user behavior or item features. The same applies in multi-label learning, where side information (e.g., feature descriptors or annotations) only partially captures label dependencies.
> - Subspace-based geometric perspective: Complete side information implies that the target matrix's column/row space is contained within the subspace spanned by the given side information. However, for datasets in our experiments, this relation does not hold. A formal check for the subset relation involves computing the projection $P_A R$ and verifying if $||P_A R - R||\_F = 0$ or close to zero, where $P_{A} = A(A^\top A)^{-1} A^\top$. A large value confirms the incomplete side information.
>
> These insights affirm that side information of the practical dataset is present but not complete.
>
> **Q4: The descreiption issue in the left column of line 426, page 8.**
>
> A4: Thank you for pointing out this confusion. We agree that the original description lacked precision. We will revise it as
> > Among them, DirtyIMC, FNNM and OCMC are applicable for matrix completion with incomplete side information.
>
> in the final version.
>
> We hope the above analysis and results sufficiently address the reviewer’s comments.

---

### Official Review · Reviewer_zbWg · 2025-03-17

**Overall Recommendation:** 3

**Summary:**

This paper addresses matrix completion with incomplete side information. The authors propose an Orthogonal Complement Matrix Completion (OCMC) model that leverages orthogonal complement projection derived from available side information. The key insight is that when side information is incomplete, focusing on the orthogonal complement projection provides valuable constraints. The authors formulate this as minimizing both the nuclear norm of the entire matrix and the nuclear norm of its orthogonal complement projection. Using PAC learning theory, they demonstrate that sample complexity decreases quadratically with the completeness level of side information. They develop a linearized Lagrangian algorithm to efficiently solve the model with convergence guarantees. Experiments on synthetic data, multi-label learning tasks, and movie recommendations show that OCMC consistently outperforms other methods.

**Claims And Evidence:**

The claims in this paper are supported by evidence:
- The claim that orthogonal complement projection plays a critical role is supported by theoretical analysis in Section 2.2 and empirical evidence in Figure 2, showing that both the rank and nuclear norm of this projection decrease with increasing side information completeness.
- The sample complexity analysis (Corollary 3.4) rigorously establishes the quadratic decrease with the completeness level.
- Performance claims are backed by comprehensive experiments across different completeness levels, observation rates, and real-world applications.

**Essential References Not Discussed:**

There are few other papers that studied the problem of matrix completion with graph side information and should be included in the literature review.
- Community detection and matrix completion with social and item similarity graphs, IEEE Transactions on Signal Processing, 2021
- The optimal sample complexity of matrix completion with hierarchical similarity graphs, ISIT 2022
- Graph-assisted matrix completion in a multi-clustered graph model, ISIT 2022
- On the fundamental limits of matrix completion: Leveraging hierarchical similarity graphs, IEEE Transaction on Information Theory, 2024

**Experimental Designs Or Analyses:**

The experimental designs are sound:
- Synthetic experiments with controlled settings allow isolation of different factors
- Real-world experiments on well-established datasets (MovieLens-100k, Yahoo web classification)
- Thorough ablation studies examining the relationship between completeness, observations, and accuracy

**Methods And Evaluation Criteria:**

The methods and evaluation criteria are appropriate for the problem:
- The OCMC formulation captures the intuition of using incomplete side information.
- The linearized ADMM algorithm addresses computational challenges in the optimization.
- Evaluation spans synthetic experiments (controlling completeness and observation rates) and real-world applications.

**Other Comments Or Suggestions:**

None.

**Other Strengths And Weaknesses:**

Strengths and weaknesses have been highlighted in other questions.

**Questions For Authors:**

1. How would you recommend setting the parameter λ in practice when the completeness level of side information is unknown? Could this be automatically determined from the data?
2. Beyond incompleteness, have you explored how noisy side information affects OCMC's performance? This seems particularly relevant for real-world applications.
3. How does OCMC's computational complexity scale with matrix dimensions and side information size? Are there approximations that could be applied to very large-scale problems?
4. Any comments on the line of research that uses neural network-based approaches to solve the same problem?

**Relation To Broader Scientific Literature:**

This work extends the existing literature on conventional matrix completion and perfect side information methodologies.

**Theoretical Claims:**

The theoretical analysis using PAC learning theory appears sound:
- The Rademacher complexity bound (Lemma 3.2) establishes the relationship with matrix dimensions and side information completeness.
- The generalization error bound (Theorem 3.3) builds on this to bound expected recovery error.
- The sample complexity result (Corollary 3.4) derives the relationship with completeness level.

---

> ### Author Rebuttal · Authors · 2025-03-31
>
> Thanks for the reviewer about the constructive questions. We have addressed all the questions point by point in the following response.
>
> **Q1: Setting of the parameter $\lambda$  when  completeness level of side information is unknown.**
>
> A1:The relation between $\lambda$ and completeness level is discussed in Section 2.2. As the completeness increases, we need to set a larger $\lambda$ to further restrict the rank of the complement projection. Since $\lambda$ is a hyper-parameter, in practice, we recommend cross-validation to fine-tune the setting of $\lambda$.
>
> **Q2: Effect of noisy side information on OCMC's performance.**
>
> A2: We agree that noisy side information can affect the performance of OCMC. However, by appropriately adjusting $\lambda$, OCMC can still be adapted to such scenarios and maintain reasonable performance.
>
> As shown in (7), the OCMC formulation is$$
> \min\_{X} ||X||\_* +\lambda ||P\_{{A}^{\perp}{B}^{\perp}}(X)||\_*
> \quad \text{s.t. } P\_{\Omega}(X)=P\_{\Omega}(R).$$
> When the side information is noisy—e.g., when matrices $A$ and $B$ are perturbed—we denote the noisy versions as $\check{A}$ and $\check{B}$. In such cases, the complement projection of target matrix $R$ may not lie close to the subspace $P_{\check{A}^{\perp}\check{B}^{\perp}}$, leading to a potentially large value of $||P_{\check{A}^{\perp}\check{B}^{\perp}}(R)||_*$.
>
> To address this issue, we suggest reducing $\lambda$ in (7) when side information becomes unreliable due to noise. In such cases, as the noise level increases, the side information becomes less helpful for matrix completion. For the extreme case when the side information is severely corrupted, we can set $\lambda=0$, and the OCMC model will reduce to the standard matrix completion:
> $$
> \min\_{X} ||X||\_* \quad
> \text{s.t. } P_{\Omega}(X)=P_{\Omega}(R).
> $$
>
> To illustrate the performance of OCMC under the noisy side information, we compared the completion error under different noise levels. In our experiment, the target matrix is a 100 $\times$ 100 rank-10 matrix, and the completeness level is 50%. The side information matrices $ A $ and $ B $ are corrupted by additive noise matrices $E_A$ and $E_B$, whose entries are i.i.d. $\mathcal{N}(0, \alpha^2/m)$ and $\mathcal{N}(0, \alpha^2/n)$, respectively.
> |Noise level|$\alpha=0$|$\alpha=0.1$|$\alpha=0.3$|$\alpha=0.5$|
> |-|-|-|-|-|
> |Observation rate=0.1|0.689|0.697|0.753|0.842|
> |Observation rate=0.15|0.515|0.523|0.601|0.751|
> |Observation rate=0.2|0.319|0.324|0.422|0.551|
>
> It can be observed that across noise levels, OCMC achieves slightly lower yet still robust MSE.
> We will include these discussions in the final submission.
>
> **Q3: Discussions about the OCMC's computational complexity.**
>
>  A3: For a target matrix $R \in \mathbb{R}^{m\times n}$  and side information $A\in\mathbb{R}^{m\times d}$, $B \in \mathbb{R}^{n\times d}$
>  (for convenience, here we assume $r_A=r_B=d$), the per-iteration complexity of the linear-ADMM algorithm for OCMC consists of three parts:
> - Calculating $P_{A^\perp B^\perp}(X): O(\min(dmn+dm^2, dmn+dn^2))$.
> - Singular Value Thresholding (SVT): $O(\min(m^2n, n^2m))$.
> - Matrix inner product: $O(mn)$.
>
> The total complexity is dominated by the SVT step, i.e., $O(\min(m^2n, n^2m))$. For large-scale problems, we can replace the full SVD in SVT with more efficient methods, such as randomized SVD and Lanczos method, whose complexities are $O(mn+r^2m+r^3)$ and $O(rmn)$, respectively, where $r$ is rank of $R$.
>
> **Q4: Some comments on the neural network-based approaches.**
>
>  A4:  We agree that neural network-based approaches, such as GCMC[1] or LightGCN [2], have been widely applied to solve similar problems in recommendation systems. These methods can achieve good performance in certain scenarios. However, they often come with high training overhead, especially when dealing with large-scale user and item sets. Moreover, due to the lack of interpretability, these models may suffer from poor generalization ability [3,4]. According to Occam's Razor, simpler models with fewer assumptions tend to generalize better. This motivates our focus on theoretically grounded, interpretable methods with lower complexity.
>
> In our future work, we also plan to explore the integration of the OCMC into neural network-based approaches, aiming to better incorporate side information and enhance the performance. These discussions will be included in our final submission.
>
> **Q5: Essential References Not Discussed.**
>
>  A5:  We are happy to include these in our final submission.
>
> We hope the above analysis and results sufficiently address the reviewer’s comments.
>
> **References**:
>
> [1] Graph Convolutional Matrix Completion. KDD, 2018.
>
> [2] Lightgcn: Simplifying and powering graph convolution network for recommendation. SIGIR, 2020.
>
> [3] Explicit factor models for explainable recommendation based on phrase-level sentiment analysis SIGIR, 2015.
>
> [4] Explainable recommendation: A survey and new perspectives. FnTIR, 2020.

---

### Decision · Program_Chairs · 2025-05-01

**Decision:**

Accept (poster)

**Comment:**

This paper investigates matrix completion with side information associated with both columns and rows. The authors introduce the Orthogonal Complement Matrix Completion model, which decomposes the target matrix into a combination of signals that can be explained by the side information and signals not explainable by them. To mitigate the ill-posedness of matrix recovery due to missing entries, the low-rank assumption is imposed, particularly crucial for recovering signals not explained by the side information. Theoretical analysis of the error bound is provided.

All reviewers voted positively on this submission. The reviews are positive about the proposed decomposition model and indicate that the claims are largely supported by theoretical and empirical evidence. However, several reviewers raised questions and concerns. Reviewer zbWg identified several missing references. Reviewer 67wE requested verification of the incompleteness of the side information in the real dataset. (The authors have acknowledged the existence of numerical evidence in the rebuttal, which should be included in the revised paper.) Reviewer posed several clarification questions, primarily concerning missing structure.

Based on these comments, the authors should revise their work accordingly. However, based on my judgement, additional review is not needed to reach an acceptance recommendation. Therefore, I recommend accepting the paper.